# The characteristics of atmospheric boundary layer height over the Arctic Ocean during MOSAiC

Shijie Peng [1], Qinghua Yang [1], Matthew D. Shupe [2,3],

Xingya Xi [1], Bo Han [1], Dake Chen [1], Sandro Dahlke [4], Changwei Liu [1*]

School of Atmospheric Sciences, Sun Yat-sen University, and Southern Marine Science and Engineering Guangdong Laboratory (Zhuhai), Zhuhai 519082, China
Cooperative Institute for Research in Environmental Sciences, University of Colorado Boulder, Boulder, CO, USA
NOAA Physical Science Laboratory, Boulder, CO, USA
Helmholtz Centre for Polar and Marine Research, Alfred Wegener Institute (AWI), Potsdam, Germany

Correspondence to: Changwei Liu (liuchw8@mail.sysu.edu.cn)

## Abstract

The important roles of the atmospheric boundary layer (ABL) in the central Arctic climate system have been recognized, but the atmospheric boundary-layer height (ABLH), defined as the layer of continuous turbulence adjacent to the surface, has rarely been investigated. Using a year-round radiosonde dataset during the Multidisciplinary drifting Observatory for the Study of Arctic Climate (MOSAiC), we improve a Richardson-number-based algorithm that takes cloud effects into consideration, and analyze the characteristics and variability of ABLH over the Arctic Ocean. The results reveal that the annual cycle is clearly characterized by a distinct peak in May and two minima in January and July. This annual variation in ABLH is primarily controlled by the evolution of ABL thermal structure. Temperature inversions in the winter and summer are intensified by seasonal radiative cooling and warm air advection with surface temperature constrained by melting, respectively, leading to the low ABLH at these times. Meteorological and turbulence variables also play a significant role in ABLH variation, including near-surface potential temperature gradient, friction velocity, and TKE dissipation rate. In addition, the MOSAiC ABLH is more suppressed than the ABLH during the Surface Heat Budget of the Arctic Ocean (SHEBA) experiment in the summer, which indicates that there is large variability in the Arctic ABL structure during summer melting season.

## 1 Introduction

In recent years, the rapidly changing climate and declining sea ice in the Arctic have been reported by numerous studies (e.g., Matveeva and Semenov, 2022; Meier and Stroeve, 2022; Esau et al., 2023). The Arctic near-surface temperature is increasing at a rate 2–3 times larger than the global average, which is referred to as Arctic amplification (Overland et al., 2019; Blunden and Arndt, 2019), and the Arctic has entered the 'new Arctic' period (Landrum and Holland, 2020). As a key component of the Arctic climate system, the atmospheric boundary layer (ABL) over the Arctic Ocean is closely associated with Arctic warming and has a big impact on sea ice loss (Francis and Hunter, 2006; Graversen et al., 2008; Wetzel and Bruemmer, 2011). Thus, it is critical to improve our understanding of Arctic ABL processes under 'new Arctic' conditions.

The ABL structure over the Arctic Ocean has unique characteristics due to the presence of semipermanent
sea ice, and is shaped by various mechanisms including interactions with the surface, free atmosphere, and
wave activity. Most studies of the Arctic ABL structure have been based on coastal observatories and limited
drifting ice stations (Knudsen et al., 2018; Vullers et al., 2021). It has been found that a predominant
temperature inversion in the lower troposphere exists in all seasons and is referred to as the "Arctic inversion"
(Andreas et al., 2000; Tjernström et al., 2009). The Arctic inversion is sometimes elevated, with regions of
near-neutral stability below the inversion (Persson et al., 2002; Tjernström et al., 2012). The Arctic vertical
structure is influenced by many factors, such as warm-air advection, surface melt, cloud-top cooling, and
turbulent mixing (Busch et al., 1982; Vihma et al., 2011; Vihma, 2014). Investigations of the ABL structure
evolution and its controlling factors are the keys to knowing the ABL's role in the Arctic atmosphere (Sterk
et al., 2014).
The atmospheric boundary-layer height (ABLH), here defined as the height of continuous turbulent
mixing extending up from the surface, is the key indicator of the ABL structure (Seibert et al., 2000; Seidel
et al., 2012). It determines the vertical extent of many atmospheric processes, such as convective transport
and aerosol distributions, and is an important parameter for weather and climate models (Holtslag et al.,
2013; Mahrt, 2014; Davy and Esau, 2016). In some previous studies, the ABLH over the Arctic Ocean is
defined as the height of the surfaced-based inversion top or the capping inversion base (e.g., Tjernström et
al., 2009; Sotiropoulou et al., 2014). However, as the most fundamental characteristic of the ABL, turbulence
is not fully considered in this definition. There are two primary layers of turbulent mixing in the Arctic
atmosphere. First, the surface layer, formed by turbulent mixing processes near the surface, is frequently
shallower than the Arctic inversion layer (Mahrt, 1981; Andreas et al., 2000). Second, the turbulence
associated with low-level clouds, which is driven by radiative cooling near the cloud top, forms a cloud-
induced mixed layer (Solomon et al., 2011; Shupe et al., 2013). This cloud-driven mixed layer is sometimes
decoupled from the surface mixed layer while at other times it extends down to form a coupled, well-mixed
layer all the way to the surface (Shupe et al., 2013; Brooks et al., 2017). Wind-shear induced turbulence can
also play a role in both of these layers and their interactions. Based on different turbulence characteristics,
the ABLH is commonly determined using profiles of potential temperature, wind speed, and humidity, and
various methods have been proposed for calculating ABLH (Seibert et al., 2000; Seidel et al., 2010).
However, the applicability of these methods in the Arctic needs to be further assessed.
Due to the lack of observations, there are few analyses of ABLH over the Arctic Ocean based on
observational data. Distributions of Arctic ABLH have been investigated by Tjernström and Graversen
(2009), Liu and Liang (2010), and Dai et al. (2011), but their studies are all based on the Surface Heat Budget
of the Arctic Ocean (SHEBA) campaign conducted 25 years ago (Uttal et al., 2002). To improve our
understanding of the ABL structure and ABLH characteristics under "new Arctic" conditions, we need new,
comprehensive observations in this environment. The Multidisciplinary drifting Observatory for the Study
of Arctic Climate (MOSAiC) expedition was, in part, designed to achieve this goal (Shupe et al., 2022).
Based on and around a drifting research vessel in the central Arctic for a whole year, the MOSAiC expedition
provided a wealth of data and related data products with unprecedented high temporal resolution and year-
round temporal coverage. These data make possible a more detailed analysis of the ABL structure evolution
and ABLH variability.
In this study, based on observational data from the MOSAiC expedition, we propose an improved ABLH
algorithm and then examine the characteristics of the ABL evolution over the 'new Arctic' sea-ice surface.
This paper is organized as follows: Section 2 briefly describes the MOSAiC expedition and the observations;
section 3 provides an ABLH determination method to evaluate several automated algorithms, and develops
an improved ABLH algorithm; section 4 presents the results of ABLH variation over the annual cycle, the
controlling factors of ABLH variation, and mechanisms of ABL development and suppression; section 5
compares the difference in ABLHs between SHEBA and MOSAiC; and conclusions are given in section 6.

## 87  2 Measurements

In this study, the SHEBA-based sounding data (Moritz, 2017) and multiple MOSAiC data are used. Here
we mainly introduce the MOSAiC expedition. The MOSAiC track is shown in Fig. 1, which is based on the
research vessel *Polarstern* (Knust, 2017), with the main period of atmospheric state observations starting in
October 2019 and ending in September 2020. *Polarstern* drifted across the central Arctic Ocean and
navigated through the sea ice north of 78° N during most of the MOSAiC year. The whole drifting period is
divided into five parts, and the vessel sailed in the gap period between some of those parts. More details are
provided in Shupe et al. (2022). The following are the descriptions of the instruments and data products used
in this paper.

### 97   2.1 Radiosonde observations and relevant data products

The radiosonde data were obtained through a partnership between the leading Alfred Wegener Institute
(AWI) , the atmospheric radiation measurement (ARM) user facility, a US Department of Energy facility
managed by the Biological and Environmental Research Program, and the German Weather Service (DWD)
(Maturilli et al., 2022). Vaisala RS41-SGP Radiosondes were regularly launched on board throughout the
whole MOSAiC year (from October 2019 to September 2020), including periods when the vessel was in
transit. The sounding frequency is normally four times per day (launched at about 5:00, 11:00, 17:00, and
23:00 UTC) and is increased to 7 times per day during periods of exceptional weather or coordination with
other observing activities. The radiosoundings provide data on the atmospheric state, including vertical
profiles of pressure, temperature, relative humidity (*RH*), and winds, from 12 m up to 30 km with a vertical
resolution of 5 m. However, the sounding data below ~100 m altitude may be contaminated by the vessel
itself. To avoid contamination affecting our analysis, we use a merged data product that combines the
soundings with measurements from a meteorological tower on the sea ice away from the vessel, and was
specifically designed to minimize ship effects and provide more reliable profiles in the lowest 100 m, which
has been recently submitted (Dahlke et al., 2023). In this paper, data quality control and a six-point moving
average in height are applied to the merged profile data to eliminate invalid data and measurement noise,
and all data are interpolated onto a regular vertical grid with 10 m intervals. In total, there are 1484 sounding
profiles available. In addition, DOE-ARM provides a Planetary Boundary Layer Height Value-Added

Product (PBLHT VAP, Riihimaki et al., 2019), which uses several different automated algorithms to compute ABLH estimates based on radiosonde profiles. This VAP provides 964 ABLH estimates, and we select 914 samples from these to ensure that the estimates obtained by all algorithms are available.

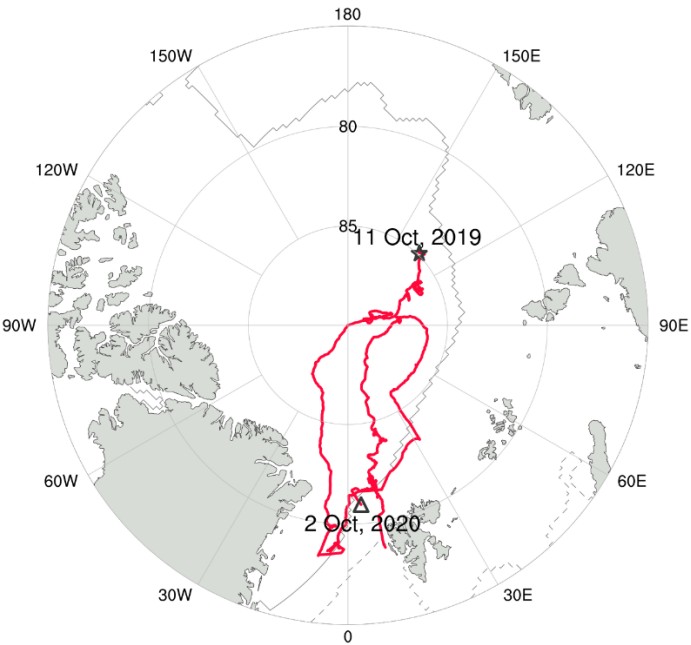

Figure 1 The MOSAiC expedition track from (star) 11 October 2019 through to (triangle) 2 October 2020 is plotted by the red line. Gray solid and dashed lines denote the approximate sea ice edge at the minimum (15 September 2020) and the maximum (5 March 2020), respectively.

### 2.2 Meteorological and turbulence measurements near the surface

Meteorological and turbulence measurements were made from a tower on the sea ice at "Met City", which was located 300–600 m away from the vessel (Cox et al., 2023). The u-Sonic-3 Cage MP anemometers by METEK GmbH and HMT300 air temperature sensors by Vaisala were fixed at nominal heights of 2 m, 6 m, and 10 m on the meteorological tower. The tower was set up during the periods when the vessel passively drifted with an ice floe (i.e., from mid-October 2019 to mid-May 2020, from mid-June through July 2020, and from late August to mid-September 2020). The sampling frequency of fast response instruments (i.e., u-Sonic-3 Cage MP anemometer) was at 20 Hz, resampled to 10 Hz. To derive turbulence parameters, the following processes were carried out: despiking, block averaging over a 10-min interval, coordinate rotating via double rotation, frequency correcting, and virtual temperature correcting. In this study, sensible heat flux (*SH*, defined as positive upwards), near-surface air temperature at 2 m, friction velocity, and turbulent kinetic energy (TKE) dissipation rate are used. Based on a footprint analysis using the Kljun et al. (2015) model, 90% of the sensible heat flux measurements have a source area fetch of no more than 275 m, a region that was typically dominated by consistent sea ice throughout the year. Although the sounding site may typically be outside the source region of these flux measurements, we assume the conditions at the two sites are equivalent, which is also assumed in the merged sounding-tower product.

## 2.3 Cloud properties derived from combined sensors

Cloud-related measurements come from ShupeTurner cloud microphysics product (Shupe, 2022). This product uses multiple measurement sources (e.g., cloud radar, ceilometer, depolarization lidar, and microwave radiometer) to derive time-height data, including cloud phase type and condensed water content for both liquid and ice. Details of the retrieval algorithm, its application, and uncertainties are provided in Shupe et al. (2015). In our study, the condensed water content data are linearly interpolated onto the vertical grid with resolution of 10 m for consistency. The cloud phase type data are used to determine clear and cloudy environments. A grid point is labeled as "cloudy" if clouds are identified in the upper and lower cloud phase type data points adjacent to the grid, otherwise it is labeled as "clear".

## 3 ABLH determination method and algorithm evaluation

The most objective method of ABLH determination is based on profiles of turbulence measurements deployed on aircraft or other platforms, but such measurements were not routinely carried out during the MOSAiC expedition. Thus, the ABLH determination in our study is based on the thermal and dynamic structure of radiosoundings. In previous literature, the ABLH is determined through multiple profiles of atmospheric variables and manual visual inspection, which can be considered as the "observed" ABLH (Liu and Liang, 2010; Zhang et al., 2014; Jozef et al., 2022). In this section, we will describe the manually-labeled ABLH determination method and derive an ABLH for each sounding. Next, we will use these ABLHs as a reference to evaluate the automated ABLH algorithms provided by the PBLHT VAP. Finally, we will develop and evaluate an improved ABLH automated algorithm that is suitable for the Arctic atmosphere, and further discuss an important parameter for the algorithms and its stability dependence.

### 3.1 ABL regime classification and ABLH determination

The ABLH determination method starts with the classification of ABL regimes. Based on previous studies (e.g., Vogelezang and Holtslag, 1996; Liu and Liang, 2010), we divide the ABLs into three types: stable boundary layer (SBL), near-neutral boundary layer (NBL), and convective boundary layer (CBL), corresponding with three different stability states near the surface. We first use $SH$ to diagnose the ABL regime types. The specific classification formula is presented below:

$$\begin{cases} SH > +\delta & \text{for CBL} \\ SH < -\delta & \text{for SBL,} \\ \text{else} & \text{for NBL} \end{cases} \quad (1)$$

where $\delta$ is the critical value that is specified as 2 W m$^{-2}$, following Steeneveld et al. (2007b). If corresponding $SH$ data are unavailable, the difference of equivalent potential temperature ($\theta_E$) between the 100 and 50 m heights ($\theta_E$ difference) derived from the sounding profile is used to determine the ABL type. Specifically, if $\theta_E$ difference is larger than 0.2 K, the ABL is identified as SBL; if $\theta_E$ difference is less than -0.2 K, the ABL is identified as CBL; and other profiles are labeled as NBLs, roughly following Liu and Liang (2010).

The manually-labeled ABLH determination in our study is based on characteristics of sounding profiles and regime types. For each atmospheric sounding profile, equivalent potential temperature ($\theta_E$), equivalent potential temperature gradient ($\theta_{Egrad}$), wind speed ($WS$), specific humidity ($q_v$), and $RH$ are used to obtain

multiple estimates of the ABLH, which are used to determine the final estimate. Three cases to describe the
method are presented in Fig. 2. Figures 2 (a–c) are the case of a SBL, which features surface-based
temperature and humidity inversions. Figures 2 (d–f) are the case of a NBL, with approximately constant $\theta_E$
from the surface up to the inversion base and strong horizontal wind. Figures 2 (g–i) are the case of a CBL,
with a deeper well-mixed layer and low-level cloud coupled to the surface (e.g., Shupe et al., 2013). In terms
of $\theta_E$ profiles, the estimated ABLH is the level at which the $\theta_{Egrad}$ reaches its maximum for SBL and NBL
cases, and the base of the $\theta_E$ inversion for CBL cases (Martucci et al., 2007). In terms of $WS$ profiles, the
ABLH is estimated to be the height of the WS maximum for all three regime types (Mahrt et al., 1979). In
terms of humidity profiles, the estimated ABLH is the level at which the $RH$ rapidly decreases for SBL and
NBL cases, and the base of the $q_v$ inversion for CBL cases (Lenschow et al., 2000).  The manually-observed
ABLHs (solid black lines in Fig. 2) are then determined through consideration of these three distinct
estimates using the following rules: (1) If the estimates differ slightly from each other, take the average of
these estimates as ABLH; (2) If a strong characteristic (sharp gradients or peaks) of the profile is evident,
select the estimate obtained based on this characteristic; (3) If the ABL structure is similar to that at the
previous time, select the estimate with the smallest change to ensure that ABLHs are consistent in time. It is
evident that the lowest layers of profiles have a great impact on the ABLH determination, particularly for
shallow SBLs and NBLs. Thus, the merged radiosonde-tower profiles help make the ABLH determination
more reliable than when using radiosondes alone.

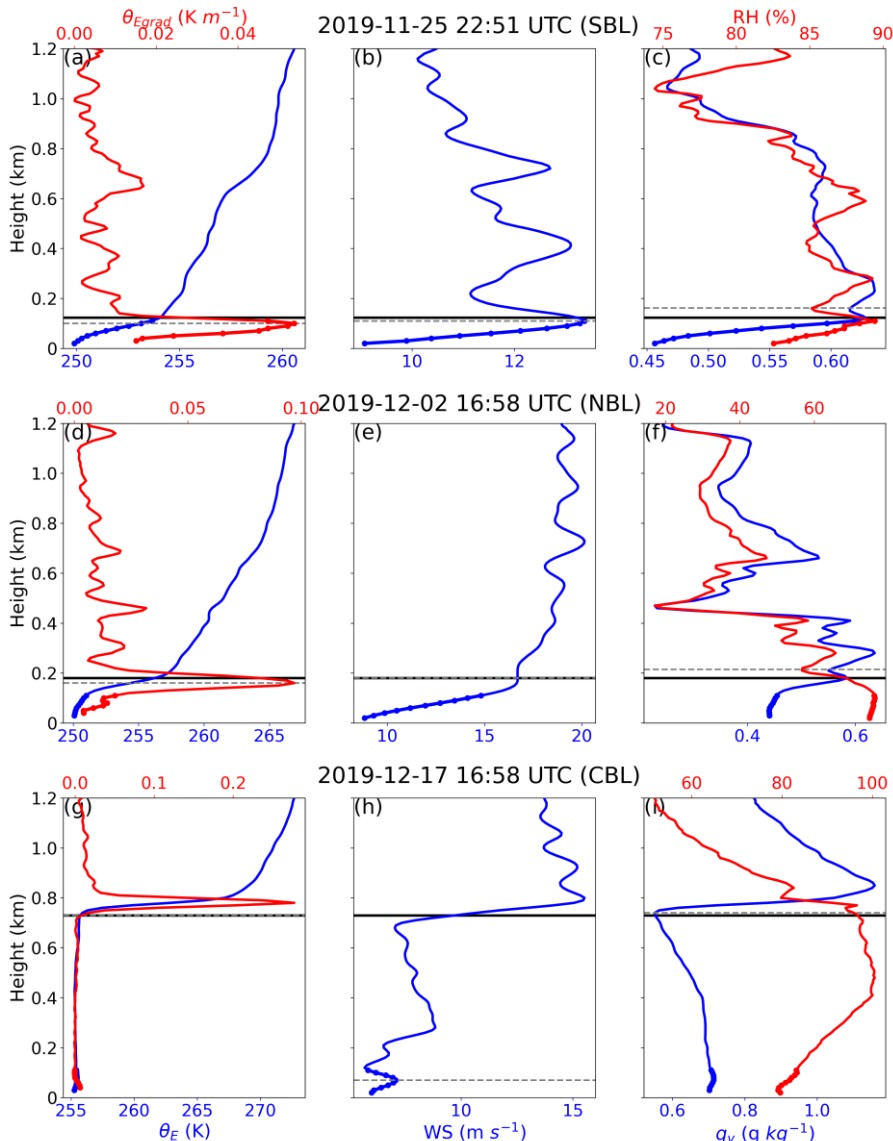

Figure 2 Vertical profiles of (left) equivalent potential temperature ($\theta_E$), $\theta_E$ gradients ($\theta_{Egrad}$), (middle) wind speed (*WS*), and (right) relative humidity (*RH*) and specific humidity ($q_v$) at (a–c) 25 November 2019, 22:51 UTC, (d–f) 2 December 2019, 16:58 UTC, and (g–i) 17 December 2019 16:58 UTC. Boundary layers at the three times represent stable boundary layer (SBL), near-neutral boundary layer (NBL), and convective boundary layer (CBL), respectively. The gray dashed horizontal lines denote the atmospheric boundary-layer height (ABLH) estimates based only on the profile shown in that panel, and the black solid horizontal lines denote the manually observed ABLHs. The dots in the lowest 100 m denote the section of the profiles impacted by the radiosonde-tower merging.

### 3.2 Automated algorithm evaluation

The automated ABLH algorithms consist of various empirical formulas. Based on these empirical formulas, estimated ABLHs are determined automatically and without manual intervention. Therefore, these algorithms can perform real-time and fast calculations on large amounts of data and are widely used in model simulations (Seibert et al., 2000; Konor et al., 2009). However, automated algorithms might lead to large

errors in estimating ABLHs, and the parameter selection in these algorithms will have a great impact on the results. In our study, estimated ABLHs obtained using three automated algorithms are compared with manually-labeled ABLHs to evaluate their performance over the Arctic Ocean. These algorithms, including the Liu-Liang algorithm, the Heffter algorithm, and the bulk Richardson number algorithm, are all available in the PBLH VAP as described in Sivaraman et al. (2013). Here we give a brief description of the three algorithms.

The Liu-Liang algorithm determines ABLH based on potential temperature and wind speed according to Liu and Liang (2010). For CBL regimes, the definition of ABLH is the height at "which an air parcel rising adiabatically from the surface becomes neutrally buoyant", and the temperature excess value is 0.1 K. For SBL regimes, two different estimates of the ABLH are obtained, if possible, based on stability criteria and wind shear criteria, respectively. For stability, the ABLH is defined as the lowest level, $k$, at which the $\theta_{Egrad}$ reaches a minimum and meets either of the following two conditions:

$$\begin{cases} \theta_{Egrad\,k} - \theta_{Egrad\,k-1} < -40 \text{ K km}^{-1} \\ \theta_{Egrad\,k+1} < 0.5 \text{ K km}^{-1}, \theta_{Egrad\,k+2} < 0.5 \text{ K km}^{-1} \end{cases}, (2)$$

where the subscripts ($k$, $k$-1, $k$+1, and $k$+2) represent the $\theta_{Egrad}$ at corresponding levels. For wind shear, the ABLH is defined as the height where the wind speed reaches a maximum that is at least 2 m s$^{-1}$ stronger than the layers immediately above and below while decreasing monotonically toward the surface (i.e., a low-level jet). The final ABLH is defined as the lower of the two heights.

The Heffter algorithm, which was suggested by Heffter (1980), is a widely used algorithm (e.g., Marsik et al., 1995; Snyder and Strawbridge, 2004). The algorithm determines ABLH through the strength of the inversion and potential temperature difference across the inversion. The ABLH is defined as the lowest layer in which the potential temperature difference between the top and bottom of the inversion is greater than 2 K. If no layer meets the criteria, the ABLH is defined as the layer at which the potential temperature gradient reaches the largest maximum.

The bulk Richardson number algorithm is based on the profile of the bulk Richardson number ($Ri_b$), and has been shown to be a reliable algorithm for determining ABLHs (Seidel et al., 2012). $Ri_b$ is a dimensionless number that represents the ratio of thermally produced turbulence to that induced by mechanical shear. The $Ri_b$ formula used in the PBLH VAP (Sørensen et al., 1998; Sivaraman et al., 2013) is expressed as:

$$Ri_b = \left(\frac{gh}{\theta_{v0}}\right)\left(\frac{\theta_{vh} - \theta_{v0}}{u_h^2 + v_h^2}\right), (3)$$

where $g$ is the acceleration of gravity; $\theta_{vh}$ and $\theta_{v0}$ are the virtual potential temperature at height $h$ and the surface, respectively; $u_h$ and $v_h$ are the horizontal wind speed component at height $h$. The ABLH is defined as the height of $Ri_b$ exceeding a critical threshold (the critical bulk Richardson number, $Ri_{bc}$; Seibert et al., 2000). The PBLH VAP includes ABLH estimates based on two widely used $Ri_{bc}$ values: 0.25 and 0.5.

To quantitatively evaluate the performance of each automatic algorithm, we introduce the correlation coefficient $R$ and two other statistical measures: the dimensionless $Bias$ and the median absolute error ($MEAE$; Steeneveld et al., 2007a). The formulas are as follows:

$$Bias = \frac{2}{n}\sum_{i=1}^{n} \frac{H_{auto} - H_{obs}}{H_{auto} + H_{obs}}, (4)$$

$$MEAE = \text{median}(|H_{auto}\text{-}H_{obs}|), (5)$$

where $H_{auto}$ is the ABLH obtained by the automated algorithm; $H_{obs}$ is the ABLH manually determined; $n$ is the number of valid sounding profile samples. According to the definitions of these statistical measures, larger $R$ and smaller $Bias$ and MEAE mean a better performance of the automated algorithm.

We also analyze the algorithm performances for cloudy and clear conditions, considering that low-level clouds containing liquid water play an important role in the Arctic ABL (Shupe and Intrieri, 2004; Brooks et al., 2017). In our study, the $RH$ threshold of 96% (Silber and Shupe, 2022) and the cloud source flag data are used for cloud detection. If a cloud is detected in the cloud source flag data and the $RH$ is larger than 96%, then the profile is labeled as cloudy. The sounding profiles that contain at least one identified cloud layer below 1500 m are classified as "cloudy", and as "clear" otherwise.

Figure 3 presents the comparisons of estimated ABLHs with the manually-labeled ABLHs, and the associated statistical measures are given in Table 1. The results show that the $Ri_b$ algorithm with $Ri_{bc}$ of 0.25 performs best overall, and particularly for SBL cases. The performance of the $Ri_b$ algorithm with $Ri_{bc}$ of 0.5 is poorer than that of the $Ri_b$ algorithm with $Ri_{bc}$ of 0.25, with overestimations of ABLHs in general, and larger errors with lower correlation coefficients for all types of ABLs. The Heffter algorithm performs well in cases of high ABLH and particularly for cloudy and CBL cases, but does significantly overestimate ABLH in a large number of cases as shown in the Fig. 3c subgraph. This is attributed to the determination criterion of the Heffter algorithm, i.e., ABLHs are determined by inversion layers, which means that large errors occur when the inversion layer is higher than the mixed layer. Additionally, while the Heffter performance in many of the ABL conditions is only marginally worse statistically than the $Ri_b$ algorithm with $Ri_{bc}$ of 0.25, its correlations are notably worse for SBL and NBL cases. The performance of the Liu-Liang algorithm is generally poorer than the other algorithms, particularly for correlation coefficient, which is probably due to the impact of noise in the lower ABLH profiles and unsuitable parameters in the algorithm. In summary, the $Ri_b$ algorithm is reliable over the Arctic Ocean and performs better than other algorithms, and this result agrees with Jozef et al. (2022). Furthermore, we will explore ways to improve the $Ri_b$ algorithm to make it more suitable for cloudy and convective conditions.

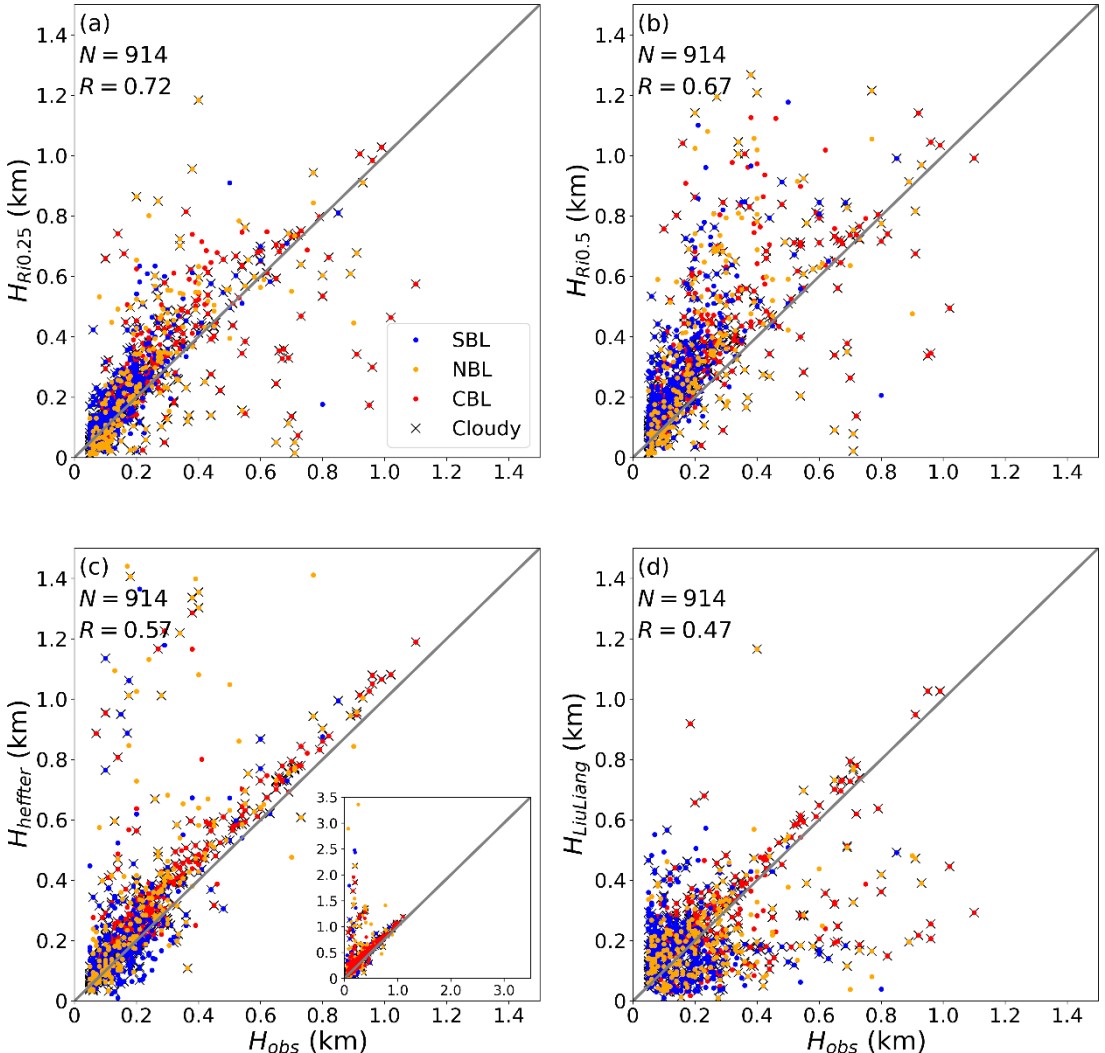

Figure 3 Comparisons of the ABLHs determined from radiosonde profiles using the bulk Richardson number
($Ri_b$) algorithm with the critical values ($Ri_{bc}$) of (a) 0.25 and (b) 0.5, (c) the Heffter algorithm, and (d) the
Liu-Liang algorithm with the manually-identified "observed" ABLHs. The blue, yellow, and red colors
indicate regime types of SBL, NBL, and CBL, respectively. The "x" signs indicate the Cloudy ABLs. The
case numbers ($N$) and correlation coefficients ($R$) are given in each panel. The subgraph in (c) denotes all
data points ranging from 0 to 3.5 km.









Table 1 The statistical measures ($R$, *Bias*, *MEAE*) for the four algorithms applied to the radiosonde dataset. All correlation coefficients are statistically significant ($p < 0.05$), except for SBL types in the Liu-Liang algorithm.

| Algorithm | Regime type | $R$ | Bias | MEAE (m) |
|---|---|---|---|---|
| The $Ri_b$ algorithm with $Ri_{bc} = 0.25$ | ALL | 0.72 | 0.10 | 50 |
| | SBL | 0.81 | 0.16 | 34 |
| | NBL | 0.68 | -0.04 | 62 |
| | CBL | 0.65 | 0.15 | 71 |
| | Cloudy | 0.69 | 0.08 | 51 |
| The $Ri_b$ algorithm with $Ri_{bc} = 0.5$ | ALL | 0.67 | 0.40 | 97 |
| | SBL | 0.73 | 0.50 | 88 |
| | NBL | 0.61 | 0.23 | 91 |
| | CBL | 0.60 | 0.39 | 120 |
| | Cloudy | 0.66 | 0.36 | 94 |
| The Heffter algorithm | ALL | 0.57 | 0.23 | 53 |
| | SBL | 0.46 | 0.17 | 33 |
| | NBL | 0.45 | 0.30 | 59 |
| | CBL | 0.66 | 0.28 | 74 |
| | Cloudy | 0.68 | 0.25 | 59 |
| The Liu-Liang algorithm | ALL | 0.47 | 0.04 | 82 |
| | SBL | 0.05 | 0.15 | 90 |
| | NBL | 0.44 | -0.07 | 81 |
| | CBL | 0.56 | -0.05 | 69 |
| | Cloudy | 0.52 | -0.01 | 82 |
| The improved $Ri$ algorithm with $Ri_{bc} = 0.35$ | ALL | 0.85 | -0.06 | 29 |
| | SBL | 0.79 | -0.08 | 21 |
| | NBL | 0.79 | -0.18 | 35 |
| | CBL | 0.87 | 0.05 | 36 |
| | Cloudy | 0.86 | -0.03 | 30 |

### 3.3 An improved $Ri$ algorithm considering the cloud effect

As a traditional $Ri_b$ formula, Eq. (3) may break down in cases of ABLs with relatively high wind speed and upper-level stratification due to the overestimation of shear production (Kim and Mahrt, 1992). Vogelezang and Holtslag (1996) proposed a finite-difference $Ri$ formula, which is expressed as:

$$Ri_F = \frac{(g/\theta_{vs})(\theta_{vh}-\theta_{vs})(h-z_s)}{(u_h-u_s)^2+(v_h-v_s)^2+bu_*^2}, \quad (6)$$

where $z_s$ is the lower boundary for the ABL, $\theta_{vs}$, $u_s$, and $v_s$ are the $\theta_v$ and wind components at the height $z_s$, respectively, $b$ is an empirical coefficient, and $u_*$ is the surface friction velocity. $Ri_F$ is considered for a parcel located somewhat above the surface to avoid the above problem, and $u_*$ is also taken into account to avoid underestimation in the situation of a uniform wind profile in the upper layer. Here, we use $Ri_F$ for clear-sky profiles and take $z_s$ and $b$ values as 40 m and 100, respectively, according to Zhang et al. (2020). As shown in Fig. 3, the estimations of cloudy ABLHs are sometimes quite poor, which motivates us to further improve the algorithm. Under cloudy conditions, the moist Richardson number ($Ri_m$) can be used to include cloud effects on the buoyancy term. Brooks et al. (2017) adopted the $Ri_m$ formula expressed as:

$$Ri_m = \frac{\left(\frac{g}{T}\right)\left(\frac{dT}{dz}+\Gamma_m\right)\left(1+\frac{Lq_s}{RT}\right) - \frac{g}{1+q_w}\frac{dq_w}{dz}}{(\frac{du}{dz})^2+(\frac{dv}{dz})^2}, \quad (7)$$

where $T$ is air temperature, $\Gamma_m$ is the moist adiabatic lapse rate, $L$ is the latent heat of vaporization, $q_s$ is the

saturation mixing ratio, and $q_w$ is the total water mixing ratio, i.e., $q_w=q_s+q_L$, where $q_L$ is the liquid water mixing ratio and is obtained based on the condensed water content. However, Eq. (6) is a gradient $Ri$ and is calculated based on local gradients of wind speed, temperature, and humidity. To be consistent with the $Ri$ formula proposed by Vogelezang and Holtslag (1996), we rewrite the formula in a finite-difference form expressed as:

$$Ri_m = \frac{\left[(g/T_s)\left(\frac{T_h-T_s}{h-z_s}+\Gamma_m\right)\left(1+\frac{Lq_{sh}}{RT_h}\right)-\frac{g}{1+q_{wh}}\frac{q_{wh}-q_{ws}}{h-z_s}\right](h-z_s)^2}{(u_h-u_s)^2+(v_h-v_s)^2+bu_*^2}, \quad (8)$$

where subscripts ($h$ and $s$) of the variables denote the calculated height, similar to Eq. (6), but note that the $s$ and $z_s$ are adjusted to 130 m, given the cloud radar blind zone. Considering that $Ri_m$ is only appropriate for the liquid-bearing cloud cases, we use the $Ri_F$ for "clear" grid points and use $Ri_m$ for "cloudy" grid cells. Using this improved approach, we evaluated the best value of $Ri_c$ to minimize the errors compared to the reference data set, arriving at an optimal value of $Ri_c=0.35$. The comparison of ABLH estimates obtained through the improved $Ri$ algorithm with the manually-labeled ABLHs demonstrates significant improvement relative to other algorithms, particularly for cloudy conditions (Fig. 4, Table 1).

Since some other studies have proposed different $Ri_c$ values for MOSAiC (e.g., Jozef et al., 2022; Barten et al., 2023; Akansu et al., 2023), we will discuss the difference in $Ri_c$ values here. The first thing to make clear is that these studies use different formulas to obtain $Ri$ profiles. Barten et al. (2023) and Akansu et al. (2023) both use the traditional $Ri_b$ algorithm based on Eq. (3), while they used $Ri_c$ values of 0.4 and 0.12, respectively. This difference was likely caused by the different methods to manually derive their reference ABLH data sets. Jozef et al. (2022) calculates the $Ri$ over a rolling 30 m altitude range, labeled as $Ri_r$, and the criterion is modified to require four consecutive data points to be above the $Ri_c$ of 0.75. In our study, we use $Ri_F$ proposed by Vogelezang and Holtslag (1996) for clear-sky conditions, and $Ri_m$ for cloudy conditions. Based on the results presented here, it is apparent that this more complex approach improves the error statistics relative to approaches based on Eq. (3). In addition, some of the differences may also related to authors using different data sets or time periods. For instance, Akansu et al. (2023) primarily used sounding data based on tether balloon for a specific sub-period of MOSAiC, and Jozef et al. (2022) used radiosondes from when they had concurrent UAV observations. The data used in our study are based on merged sounding-tower product, as mentioned above.

To further explore the differences among the four different approaches, we examine one SBL and CBL case. For a clear-sky SBL case (Fig. 5 a, b), the approaches from Akansu et al., Jozef et al. (2022), and this study all agree closely with the manual ABLH, while the Barten et al. approach results in a significant overestimation. For a cloudy-sky CBL case (Fig. 5 c, d), the approach from this study agrees with the manual ABLH, while the approach from Barten et al. overestimates the ABLH by about 30 m, and the approaches from Akansu et al. and Jozef et al. (2022) underestimate the ABLH by 130 m and 230 m, respectively. These results further demonstrate how $Ri_c$ depends on the choice of $Ri$ formula. Moreover, $Ri_c$ is not analytically derived from basic physical principles (Zilitinkevich et al. 2007), and the concept of $Ri_c$ is challenged by non-steady regimes (Zilitinkevich and Baklanov, 2002) and the hysteresis phenomenon (Banta et al., 2003;

Tjernström et al., 2009). Therefore, an objective $Ri_c$ does not exist. Rather, it is empirically used as an
algorithmic parameter to simply derive the ABLH.

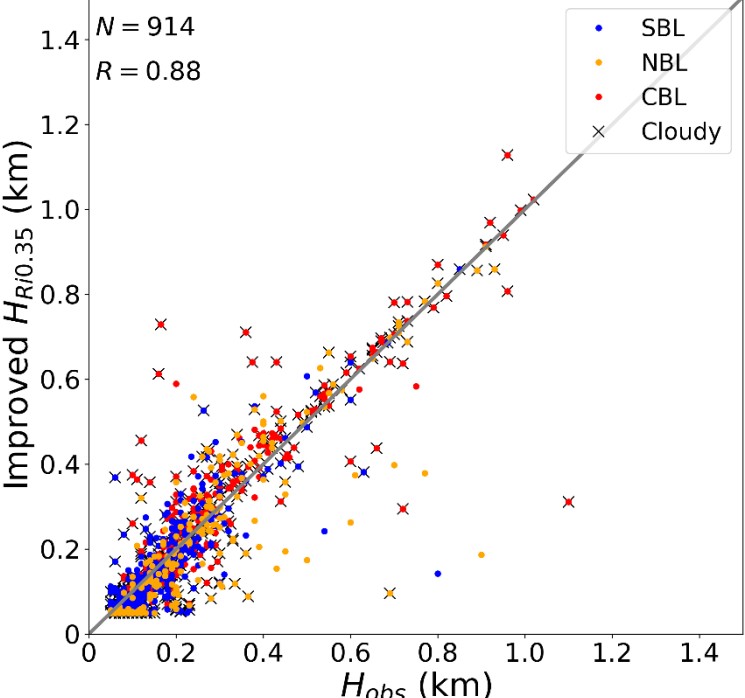


Figure 4 Similar to Fig. 3, but for the comparison of the ABLHs determined by the improved $Ri$ algorithm
with the observed ABLHs. The case number ($N$) and correlation coefficient ($R$) are given.

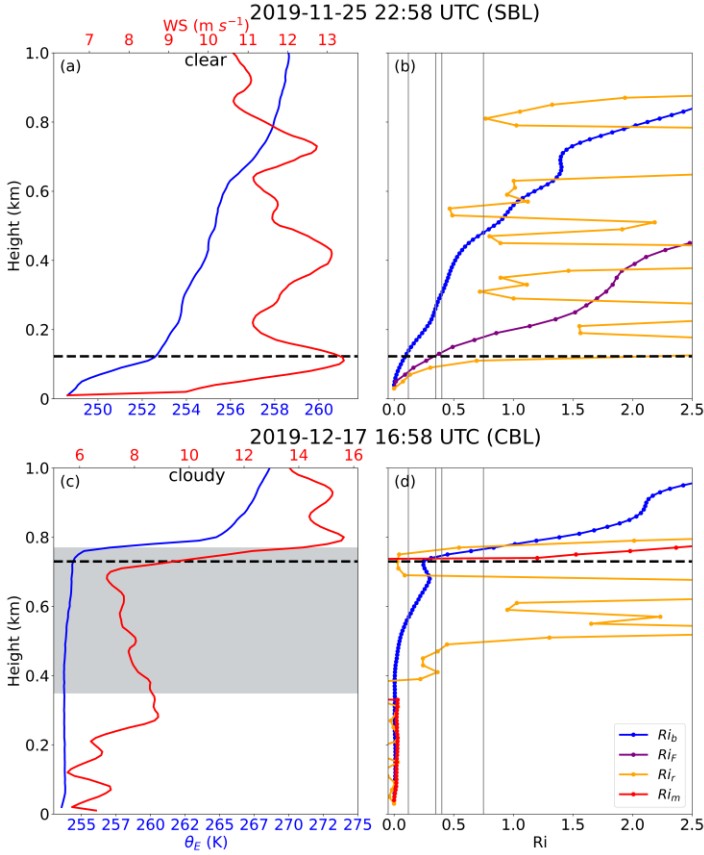


Figure 5 Vertical profiles of (left) $\theta_E$ and wind speed, and (right) $Ri$ based on different formulas at (a–b) 25
November 2019, 22:58 UTC and (c–d) 17 December 2019, 16:58 UTC. Boundary layers at the two times
represent a clear-sky SBL and a cloudy-sky CBL respectively. The black dashed horizontal lines denote the
manually-identified ABLH, and the gray solid vertical lines denote the different $Ri_c$ values, including 0.12,
0.35, 0.4, and 0.75. The gray shading in (c) denotes the cloud layer.

**3.4 The stability dependence of critical Richardson number**
Richardson et al. (2013) and Basu et al. (2014) suggested that there is a stability dependence of $Ri_c$ in
stable conditions, which is different from the constant $Ri_c = 0.35$ used in our improved algorithm. In this
section, we will discuss the impact of this dependence on ABLH estimation. We use the improved $Ri$
algorithm to calculate the $Ri$ at the manually-labeled ABLH ($h$). This new parameter is named $Ri_h$ to
distinguish it from the constant $Ri_c$. To be consistent with Basu et al. (2014), the bulk stability parameter $h/L$
is used for our analysis, where $L$ is the Obukhov length at the surface. Based on these two variables, the
stability dependence can be expressed as:
$$Ri_h = \alpha \frac{h}{L}, (9)$$

where $\alpha$ is a proportionality constant. As suggested in Basu et al. (2014), the data for convective, near-neutral,
and very stable conditions are excluded to obtain a credible $\alpha$. Specifically, data points that meet the
thresholds ($L > 500$ m and $L < L_{min}$) are excluded in our analysis, where the $L_{min}$ corresponds to the heat flux
minimum (Basu et al. 2008) and is assumed as 20 m here. Finally, we select 168 samples. The $Ri_h$ plotted as

a function of $h/L$ for these selected data is presented in Fig. 6, and the value of $L$ is colored to probe if the dependence is simply due to self-correlation. The results show $Ri_h$ values that mostly range from 0 to 0.75, and the best-fit line indicates an overall positive correlation trend, with $\alpha = 0.11$. The $\alpha$ value is somewhat larger than the results in Richardson et al. (2013) and Basu et al. (2014), which is attributed to the different $Ri$ algorithm used in our study. In addition, if a few of the extreme points are removed, the bulk of the data does not show a strong $h/L$ dependence and is instead fairly well represented by a constant $Ri_h = 0.35$, which is also suitable for convective conditions (e.g., Fig. 5c, d).

In summary, we assess the stability dependence of $Ri_c$ based on our improved $Ri$ algorithm, and the results present an overall positive correlation trend. However, this type of stability dependence of $Ri_c$ is challenged to be used in practical applications because the sensitivity of $\alpha$ to surface characteristics and atmospheric conditions can additionally degrade the accuracy of ABLH estimates. In addition, Eq. (9) requires a priori determination of the ABLH, which also causes difficulties for practical applications of such an approach. Therefore, we still use the $Ri$ algorithm with fixed $Ri_c = 0.35$ for simplicity.

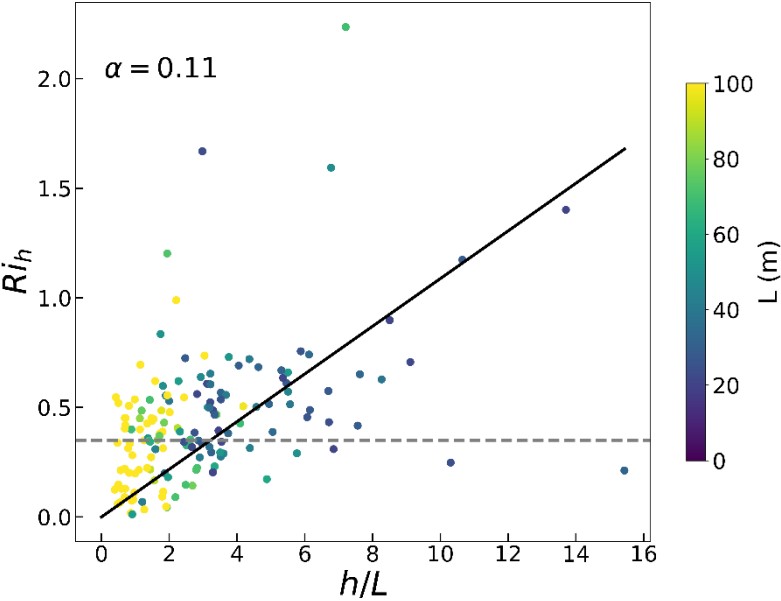

Figure 6 $Ri_h$ versus $h/L$ for selected cases. The data points are colored based on the value of $L$. The black solid line is the best fit for the selected data points, and the best-fit $\alpha$ value is also given. The gray dashed line is the constant $Ri_c = 0.35$ used in the improved $Ri$ algorithm.

## 4 MOSAiC ABLH variation and controlling factors

### 4.1 Overall distribution of ABLH

In this section, we analyze the ABLH variation during the MOSAiC and relevant controlling factors, based on the manually-labeled ABLH dataset and the ABL types that are determined through Eq. (1), or only the $\theta_E$ difference if SH is unavailable. The full-time series of ABLH during the MOSAiC expedition is presented in Fig. 7 and forms the basis for the remaining analyses. According to near surface conditions and

the sea ice state, the whole MOSAiC observation period is divided into "freeze up", "winter", "transition",
and "summer melt" periods (Shupe et al., 2022), roughly corresponding to the seasons of autumn, winter,
spring, and summer, respectively. In Figure 7, the black solid lines indicate persistent low-level clouds that
exist for more than 12 h; these occur most frequently in the late summer and autumn (the "freeze up" period),
which agrees with Shupe et al. (2011). Note that the grey dots indicate that the ABL data were observed
while the vessel was in transit, and the representativity of the ABLH data should be considered in this context.
For the first such period, the vessel left the MOSAiC ice floe in mid-May and slowly progressed south
through tightly consolidated sea ice, such that the data are generally representative of the sea ice pack in the
region. Measurements from early June when the vessel was near or in open water close to Svalbard have
been excluded entirely from the analysis. In the middle of June, as the vessel returned to the original
MOSAiC ice floe, the sea ice was not as tightly consolidated and the vessel preferentially went through leads;
the preferentially lower ice fraction along this transit could have impacted the thermal structure of the ABL.
For the three weeks in early August, the vessel moved around in the Fram Strait area and then made its way
north to another passive sea ice drifting position near the North Pole, again transiting through regions with
lower sea ice fraction. Finally, at the very end of the expedition, the vessel took some time to exit the sea ice,
stopping a few times to allow for work on the ice.
Overall, as shown in Fig. 7, the mean ABLH during the whole observation period is 231 m. This is
lower than the typical ABLH over the Arctic land surface (Liu and Liang, 2010), which is primarily attributed
to the stronger suppression of the temperature inversion over the sea-ice surface. The Arctic ABL is
suppressed for most of the MOSAiC year, while for a few periods it intensively develops for several days at
a time, most commonly when clouds and a CBL are present. For instance, frequent, intensive ABL
development occurs in the "transition" period from 13 April through to 24 May 2020. In this period, the
convective thermal structure and cloud effects contribute to ABLH reaching over the 95th percentile of the
ABLH data (horizontal dotted line) for about 7 days, with the maximum ABLH of 1100 m. In contrast, the
ABL is strongly suppressed in the period from 15 July through to 30 August 2020, with a mean ABLH of
only 136 m. The specific mechanisms of ABL development and suppression in these two cases will be
analyzed in Sections 4.3 and 4.4, respectively.
Figure 8 presents the frequency distribution of ABLH under SBL, NBL, and CBL regime types. Overall,
the sample number of SBL cases is more than that of NBL and CBL cases during the MOSAiC period (43 %
for SBL, 31% for NBL, and 26 % for CBL). These occurrence frequencies roughly agree with Jozef et al.
(2023), while their results show more NBL and CBL and less SBL. It is likely to be attributed to differences
in classification criteria. The distributions of SBL and NBL ABLH are skewed towards small values, with
94 % and 79% of the ABLH values lower than 400 m, and mean values of 165 m and 256 m, respectively.
For CBL, the distribution is shifted somewhat towards larger values, with 23 % of the ABLH values higher
than 600 m and a mean value of 309 m.

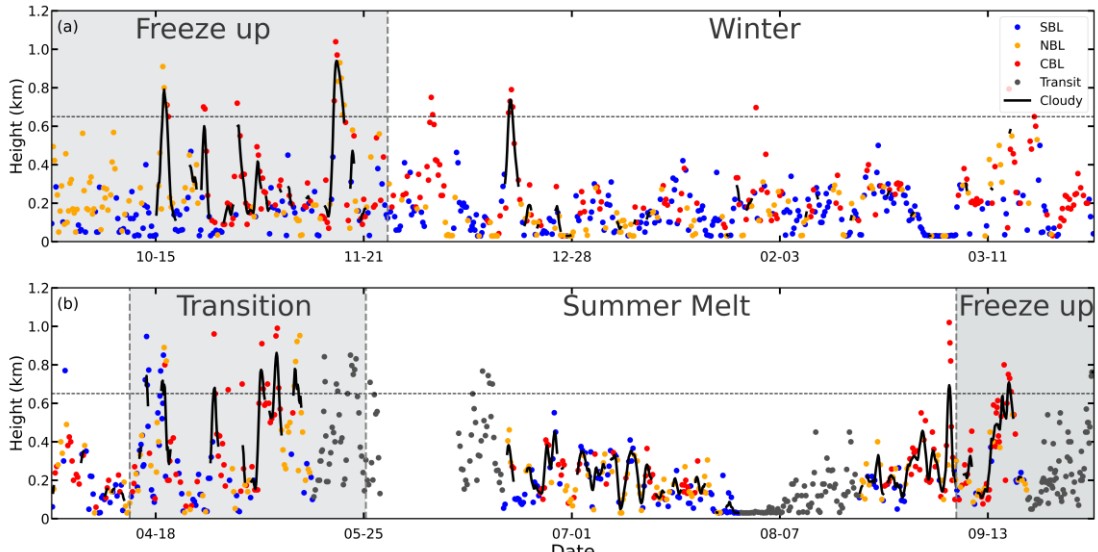

Figure 7 Time series of ABLHs throughout the MOSAiC year is divided into (a) and (b). The blue, yellow, and red dots indicate the heights of SBL, NBL, and CBL, respectively. The gray dots indicate ABL data observed while the vessel was in transit. The black solid lines indicate the heights of cloudy ABLs and persist for at least 12 hours. The gray dashed horizontal line denotes the 95th percentile of ABLH (650 m). The gray and white background shadings indicate the periods under different surface-melting states, i.e., "freeze up", "winter", "transition", and "summer melt" periods.

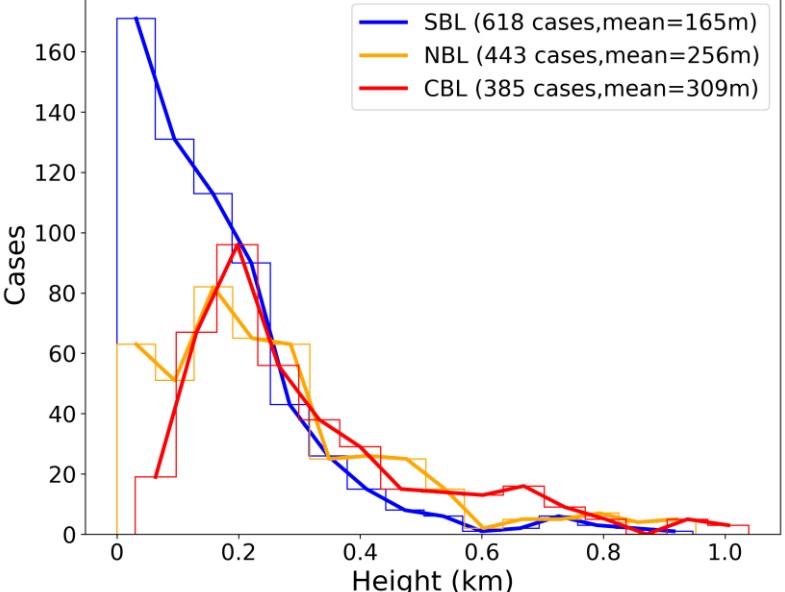

Figure 8 Frequency distribution of SBL height (blue), NBL height (yellow), and CBL height (red). The case numbers and the mean values of ABLH for SBL, NBL, and CBL conditions are also given.

## 4.2 Annual cycle of ABLH and related factors

Figure 9 presents the annual cycle of monthly ABLH statistics during the MOSAiC expedition in terms of 5th, 25th, 50th, 75th, and 95th percentiles of ABLH (boxplots) and the mean value ("x" signs and solid

and dashed lines). The box-and-whisker plots show a distinct peak in May, with a median value of 363 m and the 95th percentile reaching over 800 m. An abrupt decrease occurs in the following July and August, and another minimum occurs in January, all with median values below 150 m. It should be noted that the ABLH data in transit (gray dots in Fig. 7) are also included in the statistics, which could have been potentially impacted by more open-surface water conditions. Specifically, the ABLH data during transit periods cause higher mean ABLH for June and lower mean ABLH for August (see Fig. 7). The comparison between cloudy and clear-sky ABLHs indicates that the low-level clouds significantly contribute to the Arctic ABL development during the MOSAiC year, except in winter, when low-level clouds are rare.

The annual cycle of ABLH is determined by the seasonal evolution of the ABL structure (Tjernström et al., 2009; Palo et al., 2017), as revealed through median profiles of $\theta_E$ in each month (Fig. 10). The results show that from the start of the MOSAiC expedition (October 2019), the near-surface $\theta_E$ gradually decreases due to seasonal surface radiative cooling in the absence of sunlight, more rapidly than the atmosphere cools, which causes a strong surface temperature inversion. The increasing inversion strength through January leads to decreasing ABLH into "winter." In February and March, the surface remains steady while the atmosphere cools more, leading to diminished temperature inversion strength and a small increase in ABLH. After March 2020, with the return of sunlight, the $\theta_E$ starts to rise over the whole lower atmosphere, and the near-surface air temperature warms somewhat more than the atmosphere above. This differential warming leads to more frequent near-neutral or convective thermal structures and contributes to high ABLH during the "transition" period. In July and August, the upper-layer temperature continues to rise while the near-surface temperature is constrained to ~0 ℃ due to the melting sea ice surface, which leads again to a surface inversion and a diminished ABLH during the "summer melt" period. In September, as the sun descends to much lower angles, the $\theta_E$ across the whole lower atmosphere starts to drop, with more rapid cooling in the atmosphere relative to the near-surface resulting again in near-neutral or convective thermal structures and an increase in the CBL height during the "freeze up" period. The whole process forms these general shifts over the annual cycle. In addition, we examined the potential implications of the diurnal cycle on the ABL thermal structure. Monthly profiles based on different moments of a day were found to show little variability (not shown), such that the impact of the diurnal cycle is minimal.

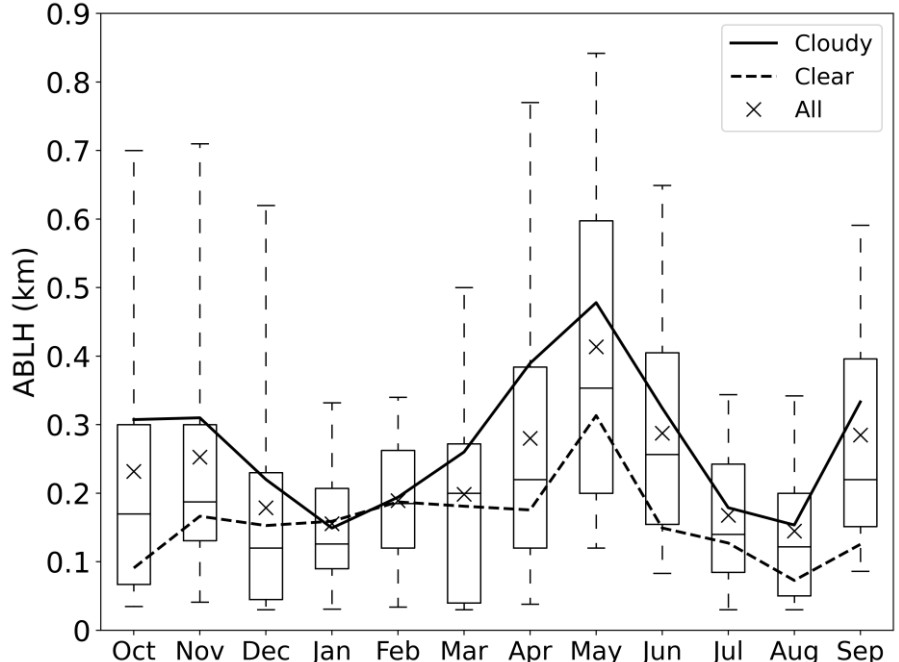


Figure 9 Box-and-whisker plots of the ABLH distribution in each month throughout the MOSAiC year. The
whiskers, the boxes, and the black horizontal lines show the 5th, 25th, 50th, 75th, and 95th percentile values
of ABLH. The solid and dashed lines and the "x" signs indicate the mean ABLH of cloudy, clear, and all
ABL types, respectively.

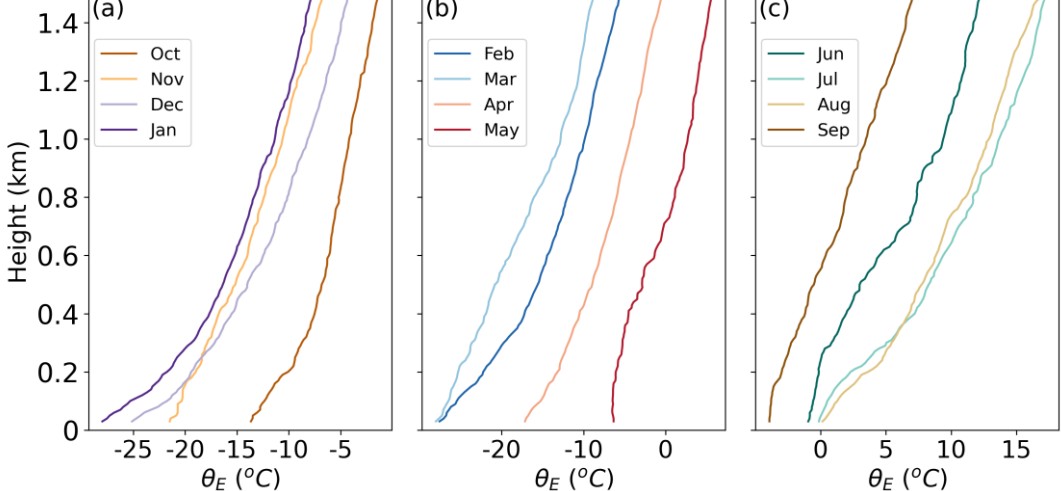


Figure 10 Monthly median profiles of equivalent potential temperature throughout the MOSAiC year. The
panel (a) represents Oct-Jan, panel (b) Feb-May and panel (c) Jun-Sep.

487         To further explore the relations between surface conditions and the ABLH, we evaluate the correlations

between the ABLH and three near-surface meteorological and turbulence parameters during the MOSAiC
period, including the near-surface equivalent potential temperature gradient ($\theta_{Egrad}=\theta_{E\ 10m}-\theta_{E\ 2m}$), friction
velocity ($u_*$), and TKE dissipation rate ($\varepsilon$). The results are shown in Fig. 11. Generally, the near-surface
buoyancy and shear effects both modulate these variables. In Fig. 11a, the ABLH distribution for negative

$\theta_{Egrad}$ has a wide range from the lowest level to above 1 km. As $\theta_{Egrad}$ becomes positive and increases, the ABLH distribution rapidly narrows to below 200 m. In general, positive $\theta_{Egrad}$ means a stably stratified ABL and surface-based temperature inversion, both of which lead to low ABLH, and negative $\theta_{Egrad}$ means that atmospheric stability near the surface is near-neutral or convective, which is necessary for ABL development. The $u_*$ presents a significant correlation with the ABLH, with correlation coefficient of 0.58 (Fig. 11b). High $u_*$ values, which are related to strong mechanical mixing, contribute to the ABL development. However, it is worth noting that intensive ABL development (ABLH over 600 m) only occurs as $u_*$ ranges between 0.2 and 0.5 m s$^{-1}$, which suggests that other factors exist to facilitate further development of the ABL, such as cloud effects (see Fig. 9). The $\varepsilon$ indicates the rate at which the TKE is changing, and the high value of $\varepsilon$ means well-developed turbulence. In Fig. 11c, when $\varepsilon$ is less than $5\times10^{-5}$ m$^2$ s$^{-3}$, turbulence in the ABL is limited with almost all ABLH values below 200 m. As $\varepsilon$ increases and becomes larger than $5\times10^{-5}$ m$^2$ s$^{-3}$, the average ABLH increases with active turbulent mixing in the ABL. The threshold of $5\times10^{-5}$ m$^2$ s$^{-3}$ is proposed by Brooks et al. (2017) as the distinction between turbulent and non-turbulent flows.

The free-flow stability (characterized by the free-flow Brunt-Väisälä frequency, $N$) can affect the ABLH (Zilitinkevich et al., 2002; Zilitinkevich and Baklanov, 2002; Zilitinkevich and Esau, 2002, 2003), and therefore is also examined here. Based on the buoyancy flux at the surface ($B_s$) and $N$, the NBLs and SBLs can be further divided into four types: the truly neutral (TN, $B_s = 0$ and $N = 0$), the conventionally neutral (CN, $B_s = 0$ and $N > 0$), the nocturnal stable (NS, $B_s < 0$ and $N = 0$), and the long-lived stable boundary layer (LS, $B_s < 0$ and $N > 0$). According to Zilitinkevich and Baklanov (2002), we calculate the $N$ and $B_s$ and reclassify the SBLs and NBLs. We find that the percentages of $N > 0.015$ in SBLs and NBLs are 89 % and 80 %, which indicates that LS and CN types dominate the stable and neutral conditions for MOSAiC, respectively. Since only 80 TN cases were identified, these are deemed to be too few for additional analysis of this type. Zilitinkevich and Esau (2003) gave ABLH equations relevant to each ABL type as:

$$h_E=\begin{cases} C_N u_* |fN|^{-1/2} & \text{(Pollard et al., 1973)} \quad \text{for CN ABL, (10)} \\ C_S u_*^2 |fB_s|^{-1/2} & \text{(Zilitinkevich, 1972)} \quad \text{for NS and LS ABL, (11)} \end{cases}$$

where $h_E$ is the equilibrium ABLH, $f$ is the Coriolis parameter, and $C_N$ and $C_S$ are empirical coefficients. In addition, Vogelezang and Holtslag (1996) and Steeneveld et al. (2007a) also explore a $h_E$ equation without taking into account $f$ explicitly, expressed as:

$$h_E=C_i \frac{u_*}{N} \quad \text{for all SBL and NBL, (12)}$$

where $C_i$ is an empirical coefficient. Here we select the CN, NS, and LS ABLH dataset, and fit the data with the corresponding expressions in Eq. (10–12) to obtain the empirical coefficients, and the results are presented in Fig. 12. All three expressions tend to well represent the ABLHs, with significant correlation coefficients. The empirical coefficients $C_N$ and $C_S$ are 1.7 and 0.4, respectively, which are close to the typical values determined through large-eddy simulations (Zilitinkevich, 2012). The coefficient $C_i = 20$ in Fig. 12c is double the typical value of 10 (Vogelezang and Holtslag, 1996), but agrees with the results reported by Overland and Davidson (1992) for the ABL over sea ice. The difference in $C_i$ may be attributed to the unique free-flow stability or other potential mechanisms of ABL development in the Arctic atmosphere.

In summary, near-surface conditions and free-flow stability play a key role in ABL development and
are also indicators, in that one can roughly determine the development state of the whole ABL from these
basic variables.

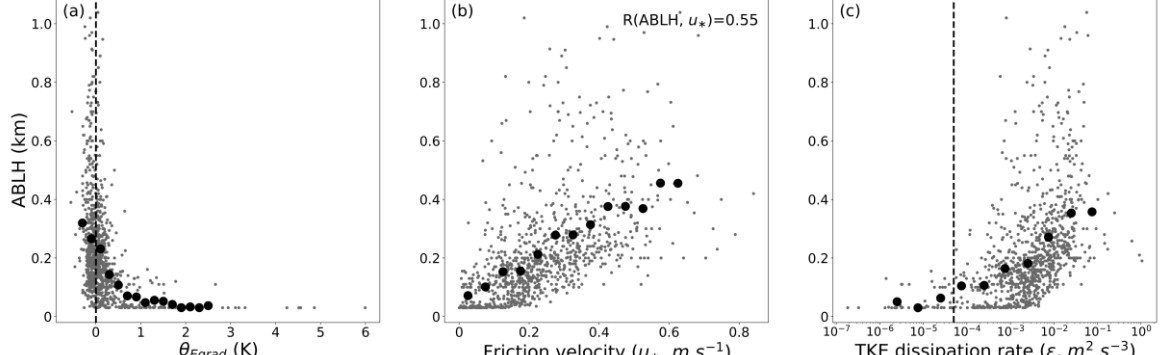


Figure 11 The ABLHs and bin-averaged values for (a) equivalent potential temperature gradient, $\theta_{Egrad}$ (K),
(b) friction velocity, $u_*$ (m s$^{-1}$), and (c) turbulent kinetic energy dissipation rate, $\varepsilon$ (m$^2$ s$^{-3}$). The average bins
for $\theta_{Egrad}$, $u_*$, and $\varepsilon$ logarithm are 0.2 K, 0.05 m s$^{-1}$, and 0.5 m$^2$ s$^{-3}$, respectively. The correlation coefficient
$R$ is given in (b), which is statistically significant ($p < 0.05$). The dashed vertical lines indicate the thresholds
of (a) $\theta_{Egrad} = 0$ K and (c) $\varepsilon = 5 \times 10^{-5}$ m$^2$ s$^{-3}$.

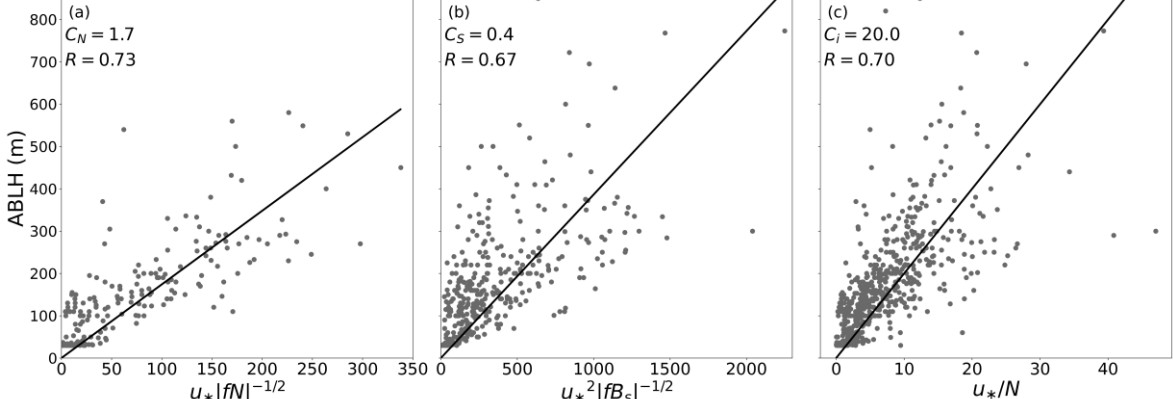


Figure 12 The ABLHs versus three expressions in Eq. (10–12). The empirical coefficients $C_N$, $C_S$, and $C_i$
are given in (a), (b), and (c), respectively, and represent the slope of the best fit line (black line). The
correlation coefficient $R$ is given in each panel, which is statistically significant ($p < 0.05$).

**4.3 Case study #1: Intensively developed ABL 13 April - 24 May 2020**
To investigate the ABL development and its controlling factors, we analyze the association of the ABLH
with vertical thermal structure and near-surface conditions during the transition period (see Fig. 7) when the
ABLH was generally the highest. Figure 13 presents time-height cross sections of $\theta_E$, wind speed, and $RH$,
and the time series of near-surface temperature and surface pressure during this period. We divide the whole
period into three parts based on the ABLH and the vertical structure of the lower troposphere. Overall, the
near-surface temperature is generally warmer than -20 °C and shows gradual warming towards the melting
point. In Period 1, a warm and moist air advection event affects the measurement area, resulting in increased
air temperature, near-saturated $RH$, strong winds throughout the lower troposphere, and low surface pressure.
The approximately constant $\theta_E$ profile near the surface facilitates exchange between the upper and lower
layers, and the high-speed wind profile enhances mechanical mixing, leading to highly developed ABL and
ABLH exceeding 600 m. In Period 2, the near-surface air temperature drops again to between -20 and -10
°C, which causes a temperature inversion and partially suppresses the ABL development. However, periodic
layers of near-saturated $RH$ extending up to 600m or more indicate the presence of clouds. The ABLH at
these times is related to the depth of the near-saturated layer, consistent with a structure where the cloud-
induced mixed layer aloft couples with the near-surface mixed layer, forming a deeper ABL and higher
ABLH (Wang et al., 2001; Shupe et al., 2013). In Period 3, a high-pressure synoptic system occurs and
suppresses the development of the ABL, but the cloud-driven turbulent mixing still exists and counteracts
the influence of the high-pressure system. In summary, the development of the ABL mainly depends on
large-scale synoptic processes, especially warm-air advection events. Additionally, the interaction between
the surface-mixed layer and cloud-mixed layer also plays a significant role in the ABL development.



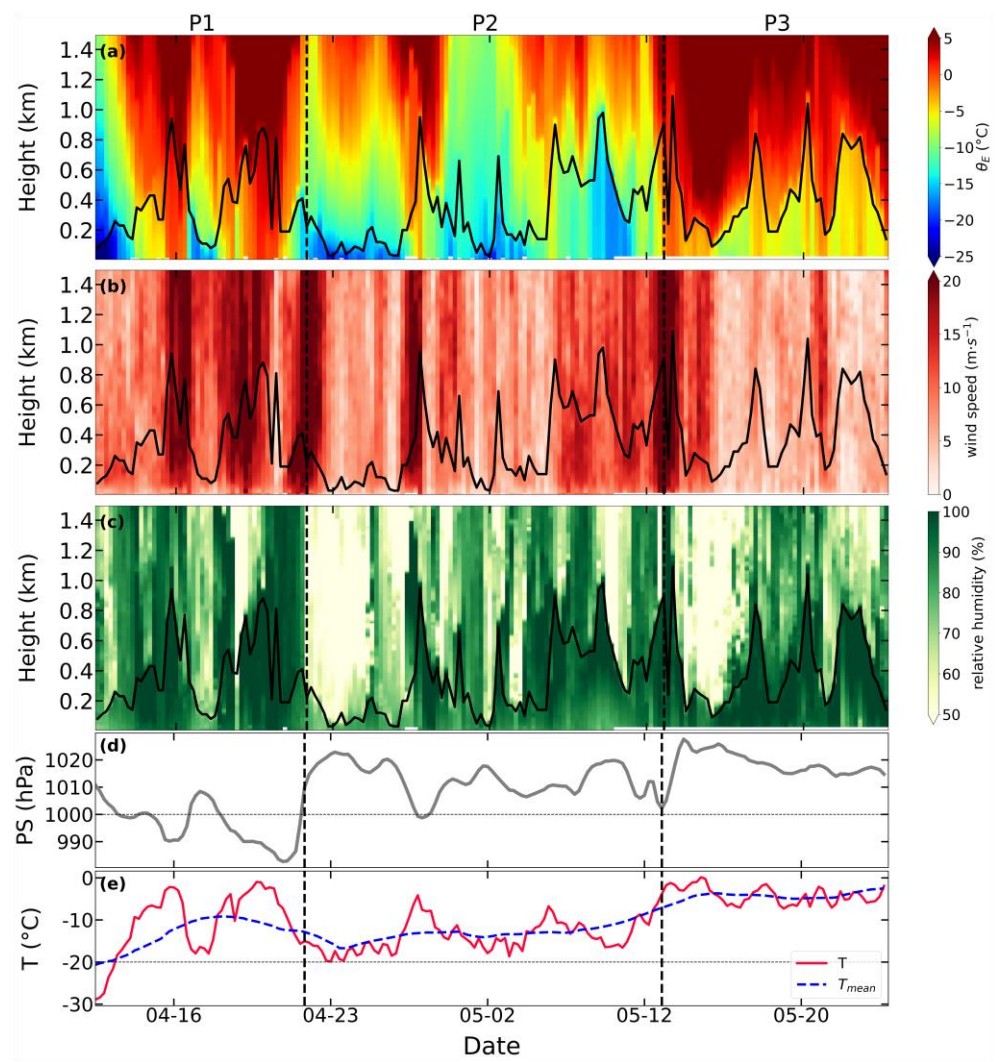

Figure 13 Time-height sections of (a) equivalent potential temperature, (b) horizontal wind speed, and (c) relative humidity and time series of (d) surface pressure and (e) near-surface air temperature (red line) and 7 d running mean of near-surface temperature (blue line). The whole period is from 13 April 2020 to 24 May 2020. Vertical dashed lines mark the identified periods P1 to P3. The black solid lines in panels (a–c) denote the ABLH during this period.

**4.4 Case study #2: the severely suppressed ABL 15 July – 30 August 2020**

The Arctic ABL is suppressed most of the time, especially in the late summer for more than a month. We choose the severely suppressed ABL in this period as a case to analyze the influences of vertical thermal structure and near-surface conditions on the ABLH. The results are shown in Fig. 14, and the whole period is divided into three parts, similar to Fig. 13. In Period 1, the near-surface air temperature is constrained to ~0 °C due to the melting surface, and the temperature inversion and weak wind are dominant throughout the lower troposphere, which suppresses the ABL development. In Period 2, warm-air advection occurs in the lower troposphere, strengthening the temperature inversion and contributing to further ABL suppression and an ABLH often lower than 100 m. Because of the constrained near-surface temperature, this structure is distinct from that of the spring "transition" period when warm-air advection facilitates ABL development.

In Period 3, the near-surface and upper-layer temperatures start to decrease, and the temperature inversion weakens, which makes the ABLH periodically grow up to ~400 m. Despite that, the ABL is still stably stratified, and the ample moisture and clouds cannot contribute significantly to the ABL development, which is consistent with Shupe et al. (2013). It is important to note that during the second half of Period 2, the *Polarstern* transited from near the sea ice edge to near the North Pole, such that this transition towards weaker temperature inversions is related to both spatial and seasonal shifts. In summary, the suppression of the ABL during the "summer melt" period results from strong temperature inversions and weak winds, and cloud-driven turbulent mixing is inhibited from interacting with the surface layer due to the near-surface stability. In this period, warm-air advection events enhance the ABL suppression, opposite to the "transition" period.

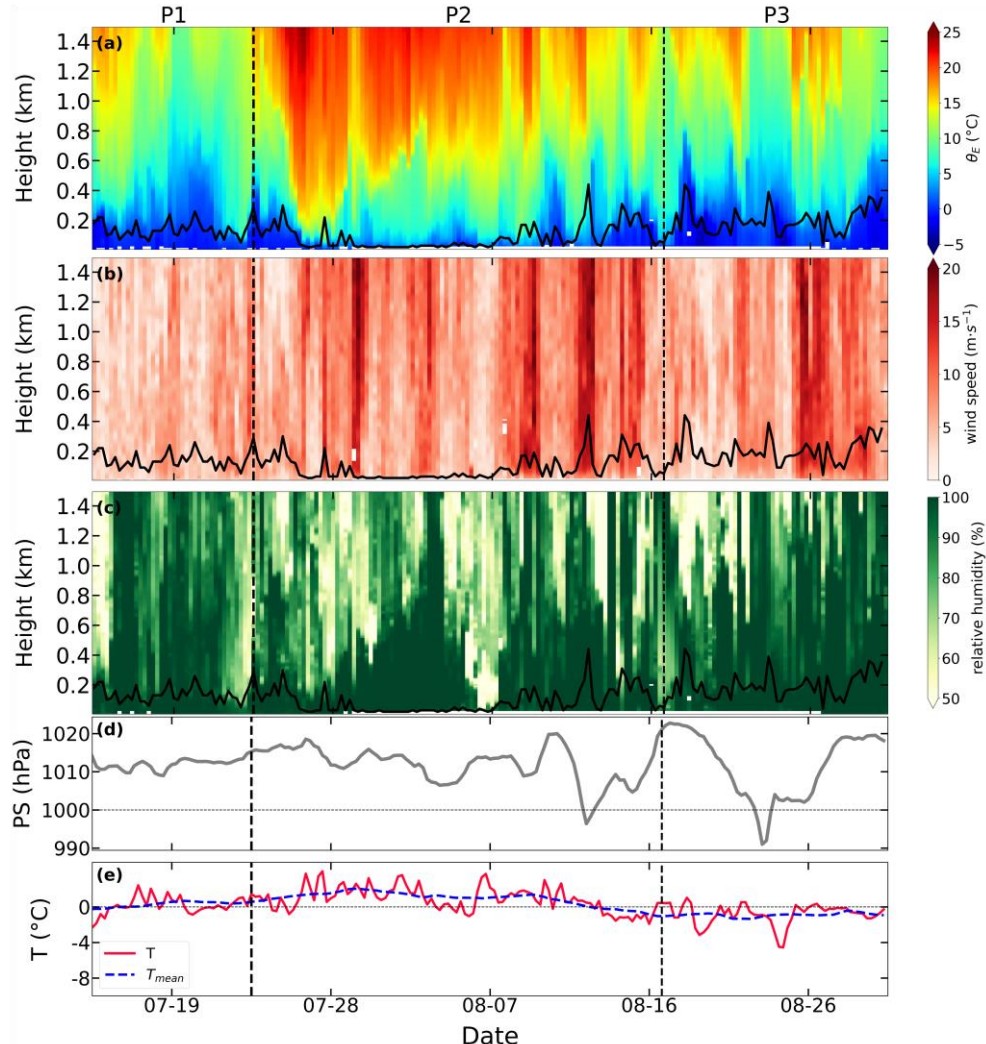

Figure 14 Similar to Fig. 13, but the period is from 15 July 2020 to 30 August 2020.

## 5 MOSAiC – SHEBA comparison

The MOSAiC and SHEBA observations were both made over the Arctic sea ice during yearlong periods. In terms of the location of observation sites, the SHEBA campaign took place in the Beaufort and Chukchi

Seas (Perovich et al., 2003), while the MOSAiC observations took place along the transpolar drift for much
of the year,in the higher latitudes of the Fram Strait in June, July, and early August, and again near the North
Pole in late August and September. The comparison between the two campaigns could provide insight into
the spatial and temporal variability in the Arctic ABL structure. The monthly ABLHs of the two campaigns
are presented in Fig. 15a. The overall distributions of ABLH are similar during the annual cycle, however,
the SHEBA ABLH is significantly higher than the MOSAiC ABLH in June and August. We will discuss
these differences based on the ABL thermal structure.
Comparisons of monthly $\theta_E$ profiles between the two campaigns during June and August are presented
in Fig. 15 (b, c). It is clear that $\theta_E$ within the lower troposphere during MOSAiC is much higher than that
during SHEBA, especially in August. In June, the near-surface $\theta_E$ values in both campaigns are close,
because both were over melting sea ice.  However, on average, the upper-layer $\theta_E$ during SHEBA is lower
than that during MOSAiC, especially at a height of around 200 m, which results in decreased low-level
stability that supports ABL development. This difference explains why the monthly SHEBA ABLH rises
from May to June, but the monthly MOSAiC ABLH decreases at this time. In July at SHEBA, the increased
air temperature in the lower troposphere combined with constrained near-surface $\theta_E$ results in a significant
temperature inversion that suppresses the ABL development (not shown). Thus, the ABLH values at SHEBA
and MOSAiC are comparable in July. In August, the $\theta_E$ profiles from the two campaigns are significantly
different. The surface at both locations is still mostly constrained to be near the melting point, while the
lower troposphere at SHEBA starts to cool more than that at MOSAiC. The SHEBA $\theta_E$ profile exhibits a
near-neutral or convective state, while the MOSAiC $\theta_E$ profile shows a further enhanced surface temperature
inversion due to warm air advection aloft, which maintains the ABL suppression. To sum up, the increase in
air temperature in the lower troposphere in early summer during MOSAiC precedes that during SHEBA,
while the cooling of the lower troposphere in late summer during MOSAiC lags that during SHEBA. These
are the main factors contributing to the ABLH differences between the two campaigns.
The atmospheric warming during the MOSAiC summer may be attributed to ongoing Arctic warming
that contributes a different atmospheric structure, but the impacts of transit periods and different synoptic
backgrounds should also be considered. First, there is the complexity of the transit periods during MOSAiC.
During the first half of June, *Polarstern* travelled northward into a somewhat loosened sea ice pack and
followed open water areas as much as possible. If anything, the higher fraction of open water along this
transit path would promote more heat exchange between the surface and ABL and higher ABLH than the
regional ice pack (e.g., Fig. 7), which suggests that the observed difference between MOSAiC and SHEBA
cannot be explained by this transit period. However, in the first part of August, when *Polarstern* transited
preferentially through open water areas during its movement further north, the transit environment was in a
persistent melting state with warm air advection aloft. It is not clear how this transit ultimately impacted the
monthly ABLH results, although the values during the transit period were lower than those during the final
10 days of August when *Polarstern* was again passively drifting with the sea ice (Figs. 7, 14). Thus, some
of the difference from SHEBA at this time could have been attributed to the specific conditions encountered
during movement of the vessel. Additionally, these two campaigns were in different storm tracks with

markedly different types of regional advection patterns. For example, in summer, MOSAiC was approaching the Fram Strait where northward warm air advection is common. Thus, synoptic variability likely plays a big role in the ABL thermal structure. In summary, there is large variability in the Arctic ABL structure during summer caused by the surface melting state, and more detailed assessments are needed to study the specific causes for the atmospheric warming and possible influences of changing Arctic conditions on the ABL structure.

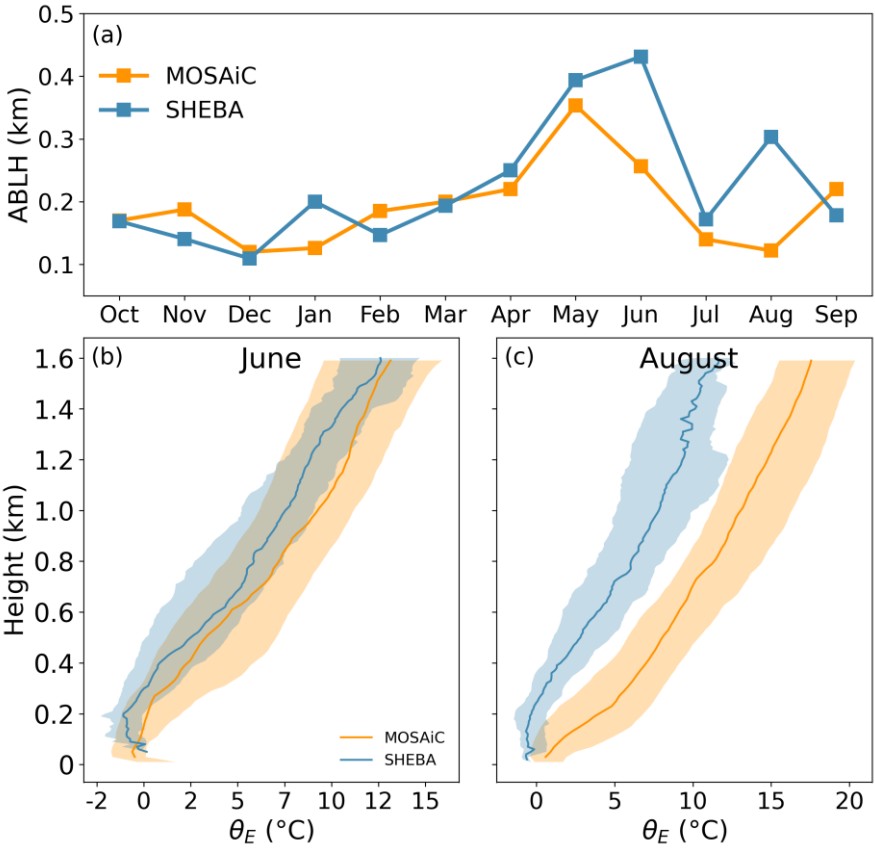

Figure 15 Comparison of ABL during SHEBA (blue squares, lines, and shadings) and MOSAiC (yellow squares, lines, and shadings), including (a) annual cycle of monthly median ABLH and monthly $\theta_E$ profiles in (b) June and (c) August. The solid lines in (b–c) indicate the median profiles, and the shadings indicate the range of 25th- and 75th- percentile profiles. The median ABLHs of SHEBA are from Dai et al. (2011).

## 6 Conclusions

This study is carried out using merged radiosounding data and corresponding surface meteorological observations and cloud properties collected during the MOSAiC expedition over a year-long period. A number of ABLH algorithms are first evaluated, prompting us to implement an improved $Ri$ algorithm that takes cloud effects into consideration. We propose a critical $Ri$ = 0.35 and further analyze its value choice and stability dependence. Subsequently, we use the manually-labeled ABLH dataset to study how atmospheric thermal structure and near-surface conditions impact the characteristics and evolution of the ABL during the MOSAiC year. Lastly, we use two cases to explore the mechanisms of ABL development and suppression over the Arctic sea-ice surface. The main conclusions are as follows.

During the MOSAiC year, the mean ABLH is 231 m, with SBLs, NBLs and CBLs accounting for 43 %, 31 %, and 26 % of the profiles, respectively. The annual cycle of the Arctic ABLH is clearly characterized by a distinct peak in May and two minima in January and July-August. Low-level clouds significantly contribute to the Arctic ABL development during the MOSAiC year, except in winter, when low-level clouds are less frequent. Compared to the SHEBA ABLH, the MOSAiC ABLH is suppressed in June and August, which is caused by increased atmospheric warming in the MOSAiC ABL during the "summer melt" period compared to SHEBA.

The annual cycle of ABLH over the Arctic Ocean is primarily controlled by the seasonal evolution of the ABL thermal structure and near-surface meteorological conditions. In the "winter" period, temperature inversions form due to negative net radiation at the surface and are associated with low ABLHs. In the spring "transition" period, the rapid increase of near-surface temperature weakens the temperature inversion, facilitating the development of the ABL. In the "summer melt" period, temperature inversions result from a fixed surface temperature at the melting point and warm-air advection aloft, which suppresses ABL development. For near-surface conditions and free-flow stability, a negative $\theta_{Egrad}$ and large TKE dissipation rate are characteristic of significant ABL development. In addition, empirical formulas relating ABLH to friction velocity, near-surface and free-flow stabilities are also tested, and the results suggest that the MOSAiC ABLH can be roughly estimated based on these basic variables.

During MOSAiC, the development of the ABL is irregular, and only occurs during intermittent periods. The year is characterized by occasions of abrupt growth of the ABLH and intensive ABLH variation for several days thereafter. These unique features are caused by large-scale synoptic processes (e.g., advection events) that bring heat, moisture, and clouds. It is worth noting that some large-scale events can have the opposite effect on the ABL. For example, warm-air advection can facilitate ABL development in the spring "transition" period but can cause ABL suppression in the "summer melt" period, when the constrained near-surface temperature cannot respond to the warmth aloft.

The findings reported here provide new insight into the annual variability and properties of the ABL and ABLH over sea ice in the 'new Arctic'. The ABLH contains information directly related to the thermal structure of the ABL and includes the impacts of weather events and large-scale circulations on the ABL structure. The ABL development supported by cloud processes was captured by the improved $Ri$ algorithm, which is similar to Brooks et al. (2017). However, the representativity of these results must still be established by comparing them with additional observations, and the influences of other variables (e.g., energy budget terms) on the ABLH should also be considered in future research.

**Data Availability**

The radiosonde data are available at the PANGAEA Data Publisher at https://doi.org/10.1594/PANGAEA.943870 (Maturilli et al., 2022). All value-added products and surface meteorological data are available at the archive of the US Department of Energy Atmospheric Radiation Measurement Program. The Planetary Boundary Layer Height Value-Added Product is available at http://dx.doi.org/10.5439/1150253 (Riihimaki et al., 2019). The cloud property data is available at https://doi.org/10.5439/1871015 (Shupe, 2022). The MOSAiC surface flux and other meteorological data are available at the Arctic Data Center at http://dx.doi.org/10.18739/A2PV6B83F (Cox et al., 2023). The

merged sounding-tower data are available at PANGAEA. The SHEBA-based sounding data are available at https://doi.org/10.5065/D6FQ9V0Z (Moritz, 2017).

## Author contribution

Shijie Peng: formal analysis; investigation; methodology; visualization; writing-original draft; writing-review and editing. Qinghua Yang: conceptualization; formal analysis; project administration; writing-review and editing; funding acquisition; supervision. Matthew D. Shupe: data curation; formal analysis; validation; resources; writing-review and editing. Xingya Xi: methodology; validation; writing-review and editing. Bo Han: formal analysis; validation; writing-review and editing. Dake Chen: formal analysis; validation; writing-review and editing; supervision. Sandro Dahlke: data curation; validation; writing-review and editing; Changwei Liu: data curation; formal analysis; validation; visualization; writing-original draft; writing-review and editing.

## Competing interests

None.

## Acknowledgments

Data used in this manuscript were produced as part of the international Multidisciplinary drifting Observatory for the Study of Arctic Climate (MOSAiC) expedition with tag MOSAiC20192020. We thank all persons involved in the expedition of the Research Vessel Polarstern during MOSAiC in 2019-2020 (AWI_PS122_00) as listed in Nixdorf et al. (2021). A subset of data was obtained from the Atmospheric Radiation Measurement (ARM) User Facility, a US Department of Energy (DOE) Office of Science User Facility Managed by the Biological and Environmental Research Program. The Alfred Wegener Institute, DOE ARM Program, and German Weather Service are acknowledged for their contributions to the MOSAiC sounding program.

## Financial Support

This study is supported by the National Natural Science Foundation of China (Nos. 42105072, 41922044, 41941009), the Guangdong Basic and Applied Basic Research Foundation (Nos. 2021A1515012209, 2020B1515020025), and the China Postdoctoral Science Foundation (Nos. 2021M693585). MDS was supported by the US National Science Foundation (OPP-1724551), the DOE Atmospheric System Research Program (DE-SC0019251, DE-SC0023036), and the National Oceanic and Atmospheric Administration Global Ocean Monitoring and Observing program (FundRef https://doi.org/10.13039/100018302, NA22OAR4320151).

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
