# Peer review of "The characteristics of atmospheric boundary layer height over the"

_EGUsphere, 2023_

## Author Comment (AC1)

Summary: In this study, the authors performed analyses on data collected during the MOSAiC campaign, focusing on the atmospheric boundary layer height (ABLH). The authors first identified the ABLH manually and then calibrated the critical bulk Richardson number in the bulk Richardson number method for computing ABLH based on the manually labeled ABLH. The relations between ABLH and surface variables were examined, and two cases were examined in detail to investigate the controlling factors of the ABLH variations during the campaign. My overall impression of the paper is that the motivation was justified, the methodology was sound, and the results made sense. I have a few comments on the bulk Richardson number method and also the language needs to be improved (beyond what I pointed out in my comments below).

Response: Thank you very much for your time and effort in reviewing our manuscript and for the constructive comments. We have revised the manuscript by addressing the referee comments, and our use of the English language has been carefully edited by all authors. The revisions in the manuscript and the reply to the comments are marked in blue.

**Major comments:**

1, the bulk Richardson number method for computing the ABLH.
1.1 Some studies also considered a friction velocity in the definition of bulk Richardson number (see e.g., Zhang et al. 2020). It might be worth discussing this.

Response: Thank you very much for your helpful advice. In our improved $Ri$ algorithm, the friction velocity ($u_*$) is now considered. The $Ri$ formula with $u_*$ can significantly improve the ABLH estimation in cases of ABLs with relatively high wind speed. In addition, we also consider the cloud effect in the improved $Ri$ algorithm to estimate ABLH better. For this comment, the corresponding changes are given in our revised manuscript as follows:

**3.3 An improved $Ri$ algorithm Considering the cloud effect**

[revised manuscript text omitted]

1.2 it is not clear whether Eq. 2 is exactly the formula used in the VAP. If so, please state it.

Response: Thank you very much for pointing this out. The Eq. (2) is exactly the formula used in the PBLH VAP. The VAP technical report cites Sivaraman et al. (2013) as a reference for the algorithms. We have added relevant statement and the reference into our revised manuscript as follows:

These algorithms, including the Liu-Liang algorithm, the Heffter algorithm, and the bulk Richardson number algorithm, are all available in the PBLH VAP, as described in Sivaraman et al. (2013).

$Ri_b$ is a dimensionless number that represents the ratio of thermally produced

turbulence to that induced by mechanical shear. The $Ri_b$ formula used in the PBLH VAP (Sørensen et al., 1998; Sivaraman et al., 2013) is expressed as:

$$Ri_b = \left(\frac{gz}{\theta_{v0}}\right)\left(\frac{\theta_{vz}-\theta_{v0}}{u_z{}^2+v_z{}^2}\right), \quad (3)$$

1.3 the authors mentioned that their results are different from Jozef et al. (2022). It would help the readers understand this by discussing a bit more of how exactly the formulations differ. Which formula did Jozef et al. use?

Response: Thank you very much for your helpful suggestions. We have added the analysis and comparison of different $Ri$ formulas and $Ri_c$ values, and the $Ri$ formula used in Jozef et al. (2022) is now also included. Jozef et al. (2022) calculates the $Ri$ over a rolling 30 m altitude range, and uses the $Ri_c$ value of 0.75. The method of calculating $Ri$ over a rolling 30 m range causes dramatic variation within the ABL, as seen in Fig. 5. Thus, for this $Ri$ definition, a large $Ri_c$ value is required to avoid the noise. The corresponding changes are given in our revised manuscript as follows:

Since some other studies have proposed different $Ri_c$ values for MOSAiC (e.g., Jozef et al., 2022; Barten et al., 2023; Akansu et al., 2023), we will discuss the difference in $Ri_c$ values here. The first thing to make clear is that these studies use different formulas to obtain $Ri$ profiles. Barten et al. (2023) and Akansu et al. (2023) both use the traditional $Ri_b$ algorithm based on Eq. (3), while they used $Ri_c$ values of 0.4 and 0.12, respectively. This difference was likely caused by the different methods to manually derive their reference ABLH data sets. Jozef et al. (2022) calculates the $Ri$ over a rolling 30 m altitude range, labeled as $Ri_r$, and the criterion is modified to require four consecutive data points to be above the $Ri_c$ of 0.75. In our study, we use $Ri_F$ proposed by Vogelezang and Holtslag (1996) for clear-sky conditions, and $Ri_m$ for cloudy conditions. Based on the results presented here, it is apparent that this more complex approach improves the error statistics relative to approaches based on Eq. (3), regardless of $Ri_c$. In addition, some of the differences may also related to authors using different data sets or time periods. For instance, Akansu et al. (2023) primarily used sounding data based on tether balloon for a specific sub-period of MOSAiC, and Jozef et al. (2022) used radiosondes from when they had concurrent UAV observations. The data used in our study are based on merged sounding-tower product, as mentioned above.

To further explore the differences among the four different $Ri$ approaches, we examine one SBL and CBL case. For a clear-sky SBL case (Fig. 5 a, b), the approaches from Akansu et al., Jozef et al. (2022), and this study all agree closely with the manual ABLH, while the Barten et al. approach results in a significant overestimation. For a cloudy-sky CBL case (Fig. 5 c d), the approach from this study agrees with the manual ABLH, while the approach from Barten et al. overestimates the ABLH by about 30 m, and the approaches from Akansu et al. and Jozef et al. (2022) underestimate the ABLH by 130 m and 230 m, respectively. These results further demonstrate how $Ri_c$ depends

on the choice of $Ri$ formula. Moreover, $Ri_c$ is not analytically derived from basic physical principles (Zilitinkevich et al. 2007), and the concept of $Ri_c$ is challenged by non-steady regimes (Zilitinkevich and Baklanov, 2002) and the hysteresis phenomenon (Banta et al., 2003; Tjernström et al., 2009). Therefore, an objective $Ri_c$ does not exist. Rather, it is empirically used as an algorithmic parameter to simply derive the ABLH.

[Figure]

Figure 5 Vertical profiles of (left) $\theta_E$ and wind speed, and (right) $Ri$ based on different formulas at (a–b) 25 November 2019, 22:58 UTC and (c–d) 17 December 2019, 16:58 UTC. Boundary layers at the two times represent a clear-sky SBL and a cloudy-sky CBL respectively. The black dashed horizontal lines denote the manually-identified ABLH, and the gray solid vertical lines denote the different $Ri_c$ values, including 0.12, 0.35, 0.4, and 0.75. The gray shading in (c) denotes the cloud layer.

2, an automated algorithm
By looking at Figure 3, why not use an automated algorithm that is based on the bulk Richardson number method for SBL and the Heffter algorithm for CBL?

Response: Thank you very much for your helpful comment. The Heffter algorithm performs well in some CBL cases, but it also causes severe ABLH overestimation. As mentioned above, we have improved the $Ri$ algorithm to better estimate ABLH in CBL cases. In addition, we realize that Fig. 3 is unclear. The data range of Fig. 3c in the

original manuscript is from 0–3.5 km due to severe ABLH overestimation by the Heffter algorithm, and thus the axis range is different from that of other panels. Therefore, we unify the axis ranges of all panels to avoid misunderstanding, and add a subgraph in Fig. 3c to denote all data points. The error statistics for different ABL regime types are also calculated and listed in Table 1. For this comment, the corresponding changes are given in our revised manuscript as follows:

(1) Changes related to Figure 3

Figure 3 presents the comparisons of estimated ABLHs with the manually-labeled ABLHs, and the associated statistical measures are given in Table 1. The results show that the $Ri_b$ algorithm with $Ri_{bc}$ of 0.25 performs best overall, and particularly for SBL cases. The performance of the $Ri_b$ algorithm with $Ri_{bc}$ of 0.5 is poorer than that of the $Ri_b$ algorithm with $Ri_{bc}$ of 0.25, with overestimations of ABLHs in general, and larger errors with lower correlation coefficients for all types of ABLs. The Heffter algorithm performs well in cases of high ABLH and particularly for cloudy and CBL cases, but does significantly overestimate ABLH in a large number of cases as shown in the Fig. 3c subgraph. This is attributed to the determination criterion of the Heffter algorithm, i.e., ABLHs are determined by inversion layers, which means that large errors occur when the inversion layer is higher than the mixed layer. Additionally, while the Heffter performance in many of the ABL conditions is only marginally worse statistically than the $Ri_b$ algorithm with $Ri_{bc}$ of 0.25, its correlations are notably worse for SBL and NBL cases. The performance of the Liu-Liang algorithm is generally poorer than the other algorithms, particularly for correlation coefficient, which is probably due to the impact of noise in the lower ABLH profiles and unsuitable parameters in the algorithm. In summary, the $Ri_b$ algorithm is reliable over the Arctic Ocean and performs better than other algorithms, and this result agrees with Jozef et al. (2022). Furthermore, we will explore ways to improve the $Ri_b$ algorithm to make it more suitable for cloudy and convective conditions.

[Figure]

Figure 3 Comparisons of the ABLHs determined from radiosonde profiles using the bulk Richardson number ($Ri_b$) algorithm with the critical values ($Ri_{bc}$) of (a) 0.25 and (b) 0.5, (c) the Heffter algorithm, and (d) the Liu-Liang algorithm with the manually-identified "observed" ABLHs. The blue, yellow, and red colors indicate regime types of SBL, NBL, and CBL, respectively. The "x" signs indicate the Cloudy ABLs. The case numbers ($N$) and correlation coefficients ($R$) are given in each panel. The subgraph in (c) denotes all data points ranging from 0 to 3.5 km.

(2) Changes related to the improved $Ri$ algorithm

**3.3 An improved $Ri$ algorithm Considering the cloud effect**

As a traditional $Ri_b$ formula, Eq. (3) may break down in cases of ABLs with relatively high wind speed and upper-level stratification due to the overestimation of shear production (Kim and Mahrt, 1992). Vogelezang and Holtslag (1996) proposed the finite-difference $Ri$ formula, which is expressed as:

$$Ri_F = \frac{(g/\theta_{vs})(\theta_{vh} - \theta_{vs})(h - z_s)}{(u_h - u_s)^2 + (v_h - v_s)^2 + bu_*^2}, (6)$$

where $z_s$ is the lower boundary for the ABL, $\theta_{vs}$, $u_s$, and $v_s$ are the $\theta_v$ and wind components at the height $z_s$, respectively, b is an empirical coefficient, and $u_*$ is the

surface friction velocity. $Ri_F$ is considered for a parcel located somewhat above the surface to avoid the above problem, and $u_*$ is also taken into account to avoid underestimation in the situation of a uniform wind profile in the upper layer. Here, we use $Ri_F$ for clear-sky profiles and take $z_s$ and b values as 40 m and 100, respectively, according to Zhang et al. (2020).

As shown in Fig. 3, the estimations of cloudy ABLHs are sometimes quite poor, which motivates us to further improve the algorithm. Under cloudy conditions, the moist Richardson number ($Ri_m$) can be used to include cloud effects on the buoyancy term. Brooks et al. (2017) adopted the $Ri_m$ formula expressed as:

$$Ri_m = \frac{\left(\frac{g}{T}\right)\left(\frac{dT}{dz} + \Gamma_m\right)\left(1 + \frac{Lq_s}{RT}\right) - \frac{g}{1 + q_w}\frac{dq_w}{dz}}{\frac{du^2}{dz} + \frac{dv^2}{dz}}, (7)$$

where $T$ is air temperature, $\Gamma_m$ is the moist adiabatic lapse rate, $L$ is the latent heat of vaporization, $q_s$ is the saturation mixing ratio, and $q_w$ is the total water mixing ratio, i.e., $q_w = q_s + q_L$, where $q_L$ is the liquid water mixing ratio and is obtained based on the condensed water content. However, Eq. (6) is a gradient Ri and is calculated based on local gradients of wind speed, temperature, and humidity. To be consistent with the $Ri$ formula proposed by Vogelezang and Holtslag (1996), we rewrite the formula in a finite-difference form expressed as:

$$Ri_m = \frac{\left[(g/T_s)\left(\frac{T_h - T_s}{h - z_s} + \Gamma_m\right)\left(1 + \frac{Lq_{sh}}{RT_h}\right) - \frac{g}{1 + q_{wh}}\frac{q_{wh} - q_{ws}}{h - z_s}\right](h - z_s)^2}{(u_h - u_s)^2 + (v_h - v_s)^2 + bu_*^2}, (8)$$

where subscripts ($h$ and $s$) of the variables denote the calculated height, similar to Eq. (6), but note that the $s$ and $z_s$ are adjusted to 130 m, given the cloud radar blind zone. Considering that $Ri_m$ is only appropriate for the liquid-bearing cloud cases, we use the $Ri_F$ for "clear" grid points and use $Ri_m$ for "cloudy" grid cells. Using this improved approach, we evaluated the best value of $Ri_c$ to minimize the errors compared to the reference data set, arriving at an optimal value of $Ri_c$=0.35. The comparison of ABLH estimates obtained through the improved $Ri$ algorithm with the manually-labeled ABLHs demonstrates significant improvement relative to other algorithms, particularly for cloudy conditions (Fig. 4, Table 1).

3, line 318:
If the manually labeled ABLH didn't include any data in transit, why did the authors think that the calibrated bulk Richardson number method using the manually labeled ABLH could be used to compute ABLH during transit?

Response: Thank you very much for your helpful comment. The data in transit contain soundings that are influenced by open ocean environment, particularly in early June. In our original manuscript, we have directly excluded the contaminated soundings from our analysis. Nonetheless, we still manually labeled the ABLHs in transit and added the relevant discussion about the in-transit period into our revised manuscript. First, we

tested the performance of the traditional and improved $Ri$ algorithms during transit periods. The results are presented in Fig. R1. It can be found that the traditional $Ri$ algorithm performs well for selected ABL cases, but with an overall overestimation. The improved $Ri$ algorithm solves this problem and shows better agreement with manually labeled ABLH. In addition, we also examined the impact of adding the manually labeled ABLH in transit into our analysis. We find that the difference in results would not affect our conclusions. Therefore, we have added the manually-labeled ABLH data during transit periods into our analysis and updated the results. The corresponding changes are given in our revised manuscript as follows:

(1) Changes related to in-transit information
The full-time series of ABLH during the MOSAiC expedition is presented in Fig. 7 and forms the bases for the remaining analyses. According to near surface conditions and the sea ice state, the whole MOSAiC observation period is divided into "freeze up", "winter", "transition", and "summer melt" periods (Shupe et al., 2022), roughly corresponding to the seasons of autumn, winter, spring, and summer, respectively. In Figure 7, the black solid lines indicate persistent low-level clouds that exist for more than 12 h; these occur most frequently in the late summer and autumn (the "freeze up" period), which agrees with Shupe et al. (2011). Note that the grey dots indicate that the ABL data were observed while the vessel was in transit, and the representativity of the ABLH data should be considered in this context. For the first such period, the vessel left the MOSAiC ice floe in mid-May and slowly progressed south through tightly consolidated sea ice, such that the data are generally representative of the sea ice pack in the region. Measurements from early June when the vessel was near or in open water close to Svalbard have been excluded entirely from the analysis. In the middle of June, as the vessel returned to the original MOSAiC ice floe, the sea ice was not as tightly consolidated and the vessel preferentially went through leads; the preferentially lower ice fraction along this transit could have impacted the thermal structure of the ABL. For the three weeks in early August, the vessel moved around in the Fram Strait area and then made its way north to another passive sea ice drifting position near the North Pole, again transiting through regions with lower sea ice fraction. Finally, at the very end of the expedition, the vessel took some time to exit the sea ice, stopping a few times to allow for work on the ice.

[Figure]

Figure 7 Time series of ABLHs throughout the MOSAiC year is divided into (a) and (b). The blue, yellow, and red dots indicate the heights of SBL, NBL, and CBL, respectively. The gray dots indicate ABL data observed while the vessel was in transit. The black solid lines indicate the heights of cloudy ABLs and persist for at least 12 hours. The gray dashed horizontal line denotes the 95th percentile of ABLH (650 m). The gray and white background shadings indicate the periods under different surface-melting states, i.e., "freeze up", "winter", "transition", and "summer melt" periods.

[Figure]

Figure R1 The Comparisons of the selected ABLHs determined by the (a) bulk Richardson number ($Ri_b$) algorithms with the critical values ($Ri_{bc}$) of 0.25 and (b) the improved $Ri$ algorithm with observed ABLHs. The blue, yellow, and red dots indicate regime types of SBL, NBL, and CBL, respectively. The case number ($N$), correlation coefficient ($R$), Bias, and median of the absolute error ($MEAE$) are given in each panel.

(2) Changes related to data selection
The sentence "In total, we select 686 samples from 964 radiosonde profiles, and all data
from observations while the vessel was in transit have been excluded" is fixed as "This
VAP provides 964 ABLH estimates, and we select 914 samples from these to ensure
that the estimates obtained by all algorithms are available", and is added into the
radiosonde description section.

**Minor comments:**

1, line 32: "has" should be "have", and add "the" before "rapid changing"

Response: Revised as suggested.

2, line 37: "and the essential place for…" can be removed.

Response: Revised as suggested.

3, line 52: add "the" before "Atmospheric boundary layer height", "referred to…"
should be removed.

Response: Revised as suggested.

4, line 56: replace "literature" with "studies"

Response: Revised as suggested.

5, line 60: replace "surface mixed layer" with "surface layer". Surface mixed layer is
odd.

Response: Revised as suggested.

6, line 107: remove "special"

Response: Revised as suggested.

7, line 108: remove "fundamental"

Response: Revised as suggested.

8, line 211: I wouldn't call this "multiple methods". Maybe change it to "multiple
profiles".

Response: Revised as suggested.

9, line 161/182/221/260: I would not call this "subjective ABLHs". Maybe "manually-labeled ABLHs".

Response: Revised as suggested.

10, line 223: replace "applied" with "available"

Response: Revised as suggested.

11, line 257: add "a" before "better performance".

Response: Revised as suggested.

12, line 302: "the smallest" should be "the best". R is not the smallest clearly.

Response: Revised as suggested.

13, line 324-327: these sentences need to be re-worded.

Response: Thank you very much for pointing this out. This statement aims to demonstrate that overall variation of the Arctic ABLH during the MOSAiC year is irregular, which is distinct from a ABL variation over land surface with diurnal cycle. We have revised these sentences as "The Arctic ABL is suppressed for most of the MOSAiC year, while for a few periods it intensively develops for several days at a time, most commonly when clouds and a CBL are present"

14, line 382: I would probably not call this "where the annual cycle began". Please reword.

Response: Thank you very much for pointing this out. We have revised the sentence as "The whole process forms these general shifts over the annual cycle."

15, line 392: how do you know a priori that it is the surface conditions that influence the ABLH, not the other way around? Please re-word.

Response: Thank you very much for pointing this out. We have revised the sentence as "To further explore the relations between surface conditions and the ABLH, we evaluate …"

16, line 397: I would not say this. The friction velocity and dissipation are affected by both shear and buoyancy.

Response: Thank you very much for pointing this out. We have revised the sentence as "Generally, the near-surface buoyancy and shear effects both modulate these variables."

17, line 408: turbulence intensity is different from turbulence kinetic energy. Do you mean turbulence intensity or turbulence kinetic energy?

Response: Thank you very much for pointing this out. We have revised it as "turbulence kinetic energy".

18, line 412: replace "accorded" with "proposed"

Response: Revised as suggested.

19, line 427: add "the" before "highest"

Response: Revised as suggested.

20, line 468: I wonder what features on the figure led the authors to conclude "the cloud-mixed layer aloft does not interact with the near-surface environment". The relative humidity is closer to saturation than figure 9 where the authors concluded "the near-saturated relative humidity indicates that the cloud-mixed layer couples with the surface-mixed layer, which facilitates the ABL development". This needs to be clarified.

Response: Thank you very much for pointing this out. We realize that this statement is unclear. Actually, as mentioned in Shupe and Intrieri (2004) and Brooks et al. (2017), the Arctic ABL structure is highly dependent on atmospheric moisture and liquid-bearing clouds. In most of the year (e.g., Fig. 9 in original manuscript), liquid-bearing clouds can create more downwelling longwave radiation and result in an ABL that is well-mixed from the surface up to the top of the saturated layer, which indicates that the cloud-mixed layer couples with the surface-mixed layer. In the Arctic summer (e.g., Fig. 10 in original manuscript), low-level stratiform clouds form as a result of ample moisture available (Tjernström et al., 2012), but the clouds cannot contribute to the ABL development due to a strong temperature inversion maintained near the surface, which is different from other seasons. For this comment, we have revised the sentence as:

"Despite that, the ABL is still stably stratified, and the ample moisture and clouds cannot contribute significantly to the ABL development, which is consistent with Shupe et al. (2013)."

21, line 556-558: it's unclear to me what the authors mean by "Coupling between the cloud mixed layer and surface mixed layer could also be recognized by the Rib

algorithm". Does the Rib method can really distinguish this?

Response: Thank you very much for pointing this out. We realize that this statement is unclear. This statement aims to demonstrate that the $Ri$ algorithm can estimate ABLH well in some cases of cloud-surface coupling, and the ABL development caused by the cloud effect can be captured by the $Ri$ algorithm. In our improved $Ri$ algorithm, the cloud effect is explicitly considered, which helps estimate ABLH better in the cloud coupling state. For this comment, we have revised the sentence as:

"The ABL development supported by cloud processes was captured by the improved $Ri$ algorithm, which is similar to Brooks et al. (2017)."

---

## Author Comment (AC2)

Summary: In this paper the authors perform an analysis of the boundary-layer height as observed from radio soundings during the MOSAiC field campaign in the Arctic in 2020. These observations are compared to PBL estimates from existing algorithms, and it is concluded that the critical Richardson number should amount to 0.15 rather than the traditional value of 0.25. The analysis has some potential, but at the same time the novelty is limited. My feeling is the paper does not build on the latest and most complete knowledge about PBL height estimation, especially not for the stable boundary layer. More can learnt from this dataset, and I find the controversial result of Ri_crit =0.15 should be discussed in more detail with findings elsewhere.

Response: Thank you very much for your time and effort in reviewing our manuscript and for the constructive comments. We have substantially revised the manuscript by addressing the comments, especially in the section that introduces and describes the PBLH algorithm. The revisions in the manuscript and the replies to the comments are marked in blue.

**Major remarks:**

1. The paper misses the opportunity to stratify the dataset of the PBL heights in more classes or groups. I.e. for example Zilitinkevich and co-workers have been working on PBL types as truly neutral PBLs, nocturnal PBLs, and conventionally neutral PBLs. In addition the analysis can be separated between profiles for cloudy/foggy vs clear sky conditions. I think this can help to reduce the scatter in Fig 3.

Response: Thank you very much for your helpful comment. We have reclassified the PBL types and added the neutral condition into our analysis. The sensible heat flux is used to determine the PBL types more credibly in our revised manuscript, which is similar to the buoyancy flux at the surface ($B_s$) applied by Zilitinkevich and co-workers. However, the truly neutral and nocturnal PBLs are not included in our PBL algorithm, because the stable and the neutral regimes are dominated by long-lived stable PBLs and conventionally neutral PBLs during the MOSAiC expedition, respectively. In addition, cloud conditions are also considered in our improved algorithm (see our response to your comment 2). For this comment, the corresponding changes are given in our revised manuscript as follows:

[revised manuscript text omitted]

2. The study misses some novelty. I understand of course that the dataset at hand is unique and very valuable, but conceptually the paper does not add much in novelty for the PBL height detection. Would it be possible to come up with a completely new

approach or PBL height formula for the PBL depth, rather than "just" retuning the Ri_crit again as was done by so many other studies before?

Response: Thank you very much for your helpful comment. We have rewritten the section on algorithm improvement. As you suggested, we have included cloudy conditions in our improved algorithm. For clear-sky conditions, we use the finite-difference $Ri$ formula proposed by Vogelezang and Holtslag (1996). While for cloudy conditions, we instead use the moist Richardson number $Ri_m$ to take the cloud effect into account, and the $Ri_m$ formula is updated based on Brooks et al. (2017). The $Ri_m$ formula used in Brooks et al. (2017) is expressed as:

$$Ri_m = \frac{\left(\frac{g}{T}\right)\left(\frac{dT}{dz} + \Gamma_m\right)\left(1 + \frac{Lq_s}{RT}\right) - \frac{g}{1 + q_w}\frac{dq_w}{dz}}{\frac{du^2}{dz} + \frac{dv^2}{dz}},$$

where $T$ is air temperature, $\Gamma_m$ is the moist adiabatic lapse rate, $L$ is the latent heat of vaporization, $q_s$ is the saturation mixing ratio, and $q_w$ is the total water mixing ratio, i.e., $q_w = q_s + q_L$, where $q_L$ is the liquid water mixing ratio.

However, it is a gradient $Ri$ and is calculated based on local gradients of wind speed, temperature, and humidity. In order to be consistent with the $Ri$ formula proposed by Vogelezang and Holtslag (1996), we rewrite the formula in a finite-difference form expressed as:

$$Ri_m = \frac{\left[(g/T_s)\left(\frac{T_h - T_s}{h - z_s} + \Gamma_m\right)\left(1 + \frac{Lq_{sh}}{RT_h}\right) - \frac{g}{1 + q_{wh}}\frac{q_{wh} - q_{ws}}{h - z_s}\right](h - z_s)^2}{(u_h - u_s)^2 + (v_h - v_s)^2 + bu_*^2},$$

which is calculated based on the difference between the height $h$ and lower reference height $z_s$. The results validate its feasibility (Figure 4). For this comment, the corresponding changes are given in our revised manuscript as follows:

(1) Changes related to cloud data description

[revised manuscript text omitted]

3. The authors have missed a paper by Barten et al. (2023) in the MOSAIC special issue in Elementa in which a similar PBL height analysis was performed. While the main focus of that paper is on the ozone budget in the Arctic PBL, it reports that the critical Richardson number should be 0.40 for MOSAIC for the same set of radio soundings. Hence this is above the typical value of 0.25, while the authors here propose 0.15. This is an obvious contradiction that needs to be discussed.

Response: Thank you very much for your helpful comment. The determination of the critical Richardson number depends on the choice of $Ri$ formula. In our original manuscript, we proposed a $Ri_c$ value of 0.15 based on a traditional $Ri_b$ formula Eq. (3). This result is different from Barten et al. (2023), who also use the Eq. (3) to propose the $Ri_c$ of 0.4. This difference may be caused by the different methods to derive the comparison ABLH dataset with which to identify the Ric. However, Barten et al. (2023) do not provide sufficient details on how their manual comparison ABLH dataset was derived. On the other hand, Akansu et al. (2023) also proposed $Ri_c = 0.12$ for MOSAiC observations based on Eq. (3), which roughly agrees with our original $Ri_c$ value of 0.15. It is worth mentioning that Akansu et al. (2023) determined the observed PBLH precisely from turbulence profiles and obtained a $Ri_c$ close to ours, which supports the reliability of our manually-labeled PBLHs. However, as mentioned in our response to your Comment 2, we have updated the Ri formula by now including cloud effects, and for that formula the new $Ri_c$ of 0.35 is identified. As you expected, this improvement reduces the scatter in our original Fig 3. For this comment, the

corresponding changes are given in our revised manuscript as follows:

Changes related to discuss difference in $Ri_c$ by previous studies

Since some other studies have proposed different $Ri_c$ values for MOSAiC (e.g., Jozef et al., 2022; Barten et al., 2023; Akansu et al., 2023), we will discuss the difference in $Ri_c$ values here. The first thing to make clear is that these studies use different formulas to obtain $Ri$ profiles. Barten et al. (2023) and Akansu et al. (2023) both use the traditional $Ri_b$ algorithm based on Eq. (3), while they used $Ri_c$ values of 0.4 and 0.12, respectively. This difference was likely caused by the different methods to manually derive their reference ABLH data sets. Jozef et al. (2022) calculates the $Ri$ over a rolling 30 m altitude range, labeled as $Ri_r$, and the criterion is modified to require four consecutive data points to be above the $Ri_c$ of 0.75. In our study, we use $Ri_F$ proposed by Vogelezang and Holtslag (1996) for clear-sky conditions, and $Ri_m$ for cloudy conditions. Based on the results presented here, it is apparent that this more complex approach improves the error statistics relative to approaches based on Eq. (3), regardless of $Ri_c$. In addition, some of the differences may also related to authors using different data sets or time periods. For instance, Akansu et al. (2023) primarily used sounding data based on tether balloon for a specific sub-period of MOSAiC, and Jozef et al. (2022) used radiosondes from when they had concurrent UAV observations. The data used in our study are based on merged sounding-tower product, as mentioned above.

To further explore the differences among the four different $Ri$ approaches, we examine one SBL and CBL case. For a clear-sky SBL case (Fig. 5 a, b), the approaches from Akansu et al., Jozef et al. (2022), and this study all agree closely with the manual ABLH, while the Barten et al. approach results in a significant overestimation. For a cloudy-sky CBL case (Fig. 5 c d), the approach from this study agrees with the manual ABLH, while the approach from Barten et al. overestimates the ABLH by about 30 m, and the approaches from Akansu et al. and Jozef et al. (2022) underestimate the ABLH by 130 m and 230 m, respectively. These results further demonstrate how $Ri_c$ depends on the choice of $Ri$ formula. Moreover, $Ri_c$ is not analytically derived from basic physical principles (Zilitinkevich et al. 2007), and the concept of $Ri_c$ is challenged by non-steady regimes (Zilitinkevich and Baklanov, 2002) and the hysteresis phenomenon (Banta et al., 2003; Tjernström et al., 2009). Therefore, an objective $Ri_c$ does not exist. Rather, it is empirically used as an algorithmic parameter to simply derive the ABLH.

[Figure]

Figure 5 Vertical profiles of (left) $\theta_E$ and wind speed, and (right) $Ri$ based on different formulas at (a–b) 25 November 2019, 22:58 UTC and (c–d) 17 December 2019, 16:58 UTC. Boundary layers at the two times represent a clear-sky SBL and a cloudy-sky CBL respectively. The black dashed horizontal lines denote the manually-identified ABLH, and the gray solid vertical lines denote the different $Ri_c$ values, including 0.12, 0.35, 0.4, and 0.75. The gray shading in (c) denotes the cloud layer.

4. The discussion section of the paper can be deepened in the sense that the Ri_crit value has been widely discussed in other papers before, but I do miss some important ones in the review, e.g. Zilitinkevich and Baklanov (2002, https://link.springer.com/article/10.1023/A:1020376832738 ). Also Basu et al (2014) proposes that the Ri_crit depends on the stability of the SBL as well (https://link.springer.com/article/10.1007/s10546-013-9878-y ). Also, Equation 2 used in the paper has been revised already by Vogelezang and Holtslag for a better score, but it feels this paper does not take benefit from this knowledge. Also, earlier LES studies for the SBL height formula are not mentioned. Hence, the current paper can be embedded more in these earlier works/contributions.

Response: Thank you very much for your helpful suggestion. We have added a discussion section to analyze the stability dependence of $Ri_c$ in stable conditions, and the results validate this relationship for the MOSAiC dataset. For improving the

algorithm that we use here, we have considered the $Ri$ formula proposed by Vogelezang and Holtslag (1996) in our ABLH algorithm, as mentioned above. The SBL height formulas based on earlier LES studies are also tested. For this comment, the corresponding changes are given in our revised manuscript as follows:

(1) Changes related to discussion on stability dependence of $Ri_c$

**3.4 The stability dependence of critical Richardson number**

Richardson et al. (2013) and Basu et al. (2014) suggested that there is a stability dependence of $Ri_c$ in stable conditions, which is different from the constant $Ri_c = 0.35$ used in our improved algorithm. In this section, we will discuss the impact of this dependence on ABLH estimation. We use the improved $Ri$ algorithm to calculate the $Ri$ at the manually-labeled ABLH ($h$). This new parameter is named $Ri_h$ to distinguish it from the constant $Ri_c$. To be consistent with Basu et al. (2014), the bulk stability parameter $h/L$ is used for our analysis, where $L$ is the Obukhov length. Based on these two variables, the stability dependence can be expressed as:

$$Ri_h = \alpha \frac{h}{L}, \quad (9)$$

[revised manuscript text omitted]

5. I was surprised that the paper never discusses whether a critical Richardson number should exist anyway. In the EFB papers by Zilitinkevich it is analytically derived that the Ri_crit does formally not exist. Though I understand that in practisal applications of Ri_crit still can have some value.

Response: Thank you very much for your helpful comment. We have added the relevant discussion in our revised manuscript. The corresponding changes are given as follows:

Since some other studies have proposed different $Ri_c$ values for MOSAiC (e.g., Jozef et al., 2022; Barten et al., 2023; Akansu et al., 2023), we will discuss the difference in $Ri_c$ values here. The first thing to make clear is that these studies use

different formulas to obtain $Ri$ profiles. Barten et al. (2023) and Akansu et al. (2023) both use the traditional $Ri_b$ algorithm based on Eq. (3), while they used $Ri_c$ values of 0.4 and 0.12, respectively. This difference was likely caused by the different methods to manually derive their reference ABLH data sets. Jozef et al. (2022) calculates the $Ri$ over a rolling 30 m altitude range, labeled as $Ri_r$, and the criterion is modified to require four consecutive data points to be above the $Ri_c$ of 0.75. In our study, we use $Ri_F$ proposed by Vogelezang and Holtslag (1996) for clear-sky conditions, and $Ri_m$ for cloudy conditions. Based on the results presented here, it is apparent that this more complex approach improves the error statistics relative to approaches based on Eq. (3), regardless of $Ri_c$. In addition, some of the differences may also related to authors using different data sets or time periods. For instance, Akansu et al. (2023) primarily used sounding data based on tether balloon for a specific sub-period of MOSAiC, and Jozef et al. (2022) used radiosondes from when they had concurrent UAV observations. The data used in our study are based on merged sounding-tower product, as mentioned above.

To further explore the differences among the four different approaches, we examine one SBL and CBL case. For a clear-sky SBL case (Fig. 5 a, b), the approaches from Akansu et al., Jozef et al. (2022), and this study all agree closely with the manual ABLH, while the Barten et al. approach results in a significant overestimation. For a cloudy-sky CBL case (Fig. 5 c d), the approach from this study agrees with the manual ABLH, while the approach from Barten et al. overestimates the ABLH by about 30 m, and the approaches from Akansu et al. and Jozef et al. (2022) underestimate the ABLH by 130 m and 230 m, respectively. These results further demonstrate how $Ri_c$ depends on the choice of $Ri$ formula. Moreover, $Ri_c$ is not analytically derived from basic physical principles (Zilitinkevich et al. 2007), and the concept of $Ri_c$ is challenged by non-steady regimes (Zilitinkevich and Baklanov, 2002) and the hysteresis phenomenon (Banta et al., 2003; Tjernström et al., 2009). Therefore, an objective $Ri_c$ does not exist. Rather, it is empirically used as an algorithmic parameter to simply derive the ABLH.

[Figure]

Figure 5 Vertical profiles of (left) $\theta_E$ and wind speed, and (right) $Ri$ based on different formulas at (a–b) 25 November 2019, 22:58 UTC and (c–d) 17 December 2019, 16:58 UTC. Boundary layers at the two times represent a clear-sky SBL and a cloudy-sky CBL respectively. The black dashed horizontal lines denote the manually-identified ABLH, and the gray solid vertical lines denote the different $Ri_c$ values, including 0.12, 0.35, 0.4, and 0.75. The gray shading in (c) denotes the cloud layer.

**Minor remarks:**

Ln 14: hyphenation: boundary layer height -> boundary-layer height. Please check whole document.

Response: Revised as suggested.

Ln 17: perhaps it is good to mention in then abstract before coming up with a new RI_crit how you defined the ABLH in your study. I.e. the level of the largest d_theta/dz, the backscatter level of a ceilometer, the low-level jet height, etc etc.?

Response: Thank you very much for your helpful suggestion. The ABLH in our study

is defined as the layer of continuous turbulence adjacent to the surface. Profiles of equivalent potential temperature, wind speed, and humidity are used in the manual ABLH determination method. The corresponding change is given in our revised manuscript as follows:

The important roles of the atmospheric boundary layer (ABL) over the Arctic Ocean in the Arctic climate system have been recognized, but the atmospheric boundary-layer height (ABLH), defined as the layer of continuous turbulence adjacent to the surface, has rarely been investigated.

Ln 33: Kwok, 2018; Hartfield et al., 2018. I have nothing against these studies but are they still recent?

Response: Thank you very much for your helpful comment. We have introduced the recent studies. The corresponding change is given in our revised manuscript as follows:

In recent years, the rapidly changing climate and declining sea ice in the Arctic have been reported by numerous studies (e.g., Matveeva and Semenov, 2022; Meier and Stroeve, 2022; Esau et al., 2023).

Ln 42: "various mechanisms and interactions with the surface": I would say the opposite since turbulent fluxes in the Arctic are usually small so the interaction with the surface is small. In the hierarchy of PBL types by Zilitinkevich et al the Arctic PBL height is characterised as a long-lived stable boundary layer where the PBL height scales more with the stratification in the free atmosphere (and wave activity therein) than with the fluxes at the surface.

Response: Thank you very much for pointing this out. We have revised the sentence as "The ABL structure over the Arctic Ocean has unique characteristics due to the presence of semipermanent sea ice, and is shaped by various mechanisms including the interactions with the surface, free atmosphere and wave activity."

Ln 51: The study by Sterk et al (2014) nicely summarizes this (https://doi.org/10.1002/jgrd.50158).

Response: Thank you very much for pointing this out. We have introduced this study. The corresponding change is given in our revised manuscript as follows:

Investigations of the ABL structure evolution and its controlling factors are the keys to knowing the ABL's role in the Arctic atmosphere (Sterk et al., 2014).

Ln 56: There are many more recent studies that indicate this as well than Deardorff,

1972; Suarez et al., 1983; Holtslag and Nieuwstadt, 1986. Please connect to the recent work!

Response: Thank you very much for your suggestion. We have introduced the recent studies. The corresponding change is given in our revised manuscript as follows:

… and is an important parameter for weather and climate models (Holtslag et al., 2013; Mahrt, 2014; Davy and Esau, 2016).

Ln 109: over the altitude range of 12 m up to 30 km. Please add what is the typical vertical resolution of the sounding measurements in the profile near the surface, this is important to know to what extent the ABLH can be well estimated.

Response: Thank you very much for your helpful comment. According to the description of the data, the radiosondes ascend at a rate of approximately 5 m s$^{-1}$, sampling with a frequency of 1 Hz, which indicates that the vertical resolution of the sounding measurements is 5 m. We have added the information into the revised manuscript as follows:

The radiosoundings provide data on the atmospheric state, including vertical profiles of pressure, temperature, relative humidity (*RH*), and winds, from 12 m up to 30 km with the vertical resolution of 5 m.

Ln 116: Moreover, we cut off the sounding data observed below 100 m altitude considering the potential contamination of the vessel itself. Please add how many of the launches had to be excluded because of the restriction.
Ln 116: Moreover, we cut off the sounding data observed below 100 m altitude considering the potential contamination of the vessel itself. The ABLH is typically shallow in the Arctic, so is the part that is eliminated not exactly the part you are interested in.
Ln 116: the section should finish with a statement how many soundings are available for analysis after all the correction and control exercises.

Response: Thank you very much for your helpful comment. When we neglect data below 100m this does have an impact on the ABLH determination, particularly for shallow SBLs. Therefore, we replaced the original data with a new merged sounding dataset, which combines the soundings with the meteorological tower data on the sea ice (Dahlke et al., 2023) with the specific goal of correcting for ship effects and providing more reliable profiles in the lowest 100 m. This new dataset allowed us to now use 1484 sounding profiles available. We updated the results and found significant improvements in SBL height determination and estimation. Also, we removed the high-resolution sounding data and used this merged data in ABLH variation section for consistency. The relevant statement is added into our revised manuscript as follows:

(1) Changes related to data description

**2.1 Radiosonde observations and relevant data products**

The radiosonde data were obtained through a partnership between the leading Alfred Wegener Institute (AWI) , the atmospheric radiation measurement (ARM) user facility, a US Department of Energy facility managed by the Biological and Environmental Research Program, and the German Weather Service (DWD) (Maturilli et al., 2022). Vaisala RS41-SGP Radiosondes were regularly launched on board throughout the whole MOSAiC year (from October 2019 to September 2020), including periods when the vessel was in transit. The sounding frequency is normally four times per day (launched at about 5:00, 11:00, 17:00, and 23:00 UTC) and is increased to 7 times per day during periods of exceptional weather or coordination with other observing activities. The radiosoundings provide data on the atmospheric state, including vertical profiles of pressure, temperature, relative humidity ($RH$), and winds, from 12 m up to 30 km with a vertical resolution of 5 m. However, the sounding data below ~100 m altitude may be contaminated by the vessel itself. To avoid contamination affecting our analysis, we use a merged data product that combines the soundings with measurements from a meteorological tower on the sea ice away from the vessel, and was specifically designed to minimize ship effects and provide more reliable profiles in the lowest 100 m, which has been recently submitted (Dahlke et al., 2023). In this paper, data quality control and a six-point moving average in height are applied to the merged profile data to eliminate invalid data and measurement noise, and all data are interpolated onto a regular vertical grid with 10 m intervals. In total, there are 1484 sounding profiles available. In addition, DOE-ARM provides a Planetary Boundary Layer Height Value-Added Product (PBLHT VAP, Riihimaki et al., 2019), which uses several different automated algorithms to compute ABLH estimates based on radiosonde profiles. This VAP provides 964 ABLH estimates, and we select 914 samples from these to ensure that the estimates obtained by all algorithms are available.

(2) Changes related to ABLH determination
It is evident that the lowest layers of profiles have a great impact on the ABLH determination, particularly for shallow SBLs and NBLs. Thus, the merged radiosonde-tower profiles help make the ABLH determination more reliable than when using radiosondes alone.

[Figure]

Figure 2 Vertical profiles of (left) equivalent potential temperature ($\theta_E$), $\theta_E$ gradients ($\theta_{Egrad}$), (middle) wind speed (*WS*), and (right) relative humidity (*RH*) and specific humidity ($q_v$) at (a–c) 25 November 2019, 22:51 UTC, (d–f) 2 December 2019, 16:58 UTC, and (g–i) 17 December 2019 16:58 UTC. Boundary layers at the three times represent stable boundary layer (SBL), near-neutral boundary layer (NBL), and convective boundary layer (CBL), respectively. The gray dashed horizontal lines denote the atmospheric boundary-layer height (ABLH) estimates based on multiple profiles, and the black solid horizontal lines denote the manually observed ABLHs. The dots at the lowest 100 m altitude denote the merged profiles.

Section 2.3: The authors should explain in more detail what is the size of the footprint of these fluxes, and to what extent they are expected to relate to the ABLH.

Response: Thanks for pointing this out. The footprint analysis by using the Kljun and others (2015) model indicates that 90% of the measured flux is expected to come from

within 275 m of the eddy-covariance observation site in average, and this fetch is characterized by sea-ice surface. The sounding site is located 300–600 m away from the meteorological tower. Although the sounding launch site is not often within the source region of the flux measurements, we assume that the spatial variation of turbulent fluxes within a kilometer range over the local sea-ice surface can be ignored.

The following description has been added into our revised manuscript:

The sentence "We neglect the distance between the vessel and 'Met City' and consider that their ABL conditions are the same, particularly when considered on hourly timescales" has been revised as: "Based on a footprint analysis using the Kljun et al. (2015) model, 90% of the sensible heat flux measurements have a source area fetch of no more than 275 m, a region that was typically strongly dominated by consistent sea ice throughout the year. Although the sounding site may typically be outside the source region of these flux measurements, we assume the conditions at the two sites are predominantly equivalent, which is also assumed in the merged sounding-tower product."

Ln 169: please add more justification why 2 classes of ABLH types are sufficient. The SBL part was earlier subdivided by many studies by Zilitinkevich in the truly neutral PBL, the nocturnal SBL and the long-lived PBL. These concepts may help to further explain the observations.

Response: As mentioned above, we have added the NBL type into our analysis, and additionally separated the ABLs into cloudy and clear conditions. Additionally, we now analyze the subdivisions of TN, CN, NS, and LS types in the correlation analysis section. The corresponding changes are given in our revised manuscript as follows:

(1) Changes related to regime classification
**3.1 ABL regime classification and ABLH determination**
The ABLH determination method starts with the classification of ABL regimes. Based on previous studies (e.g., Vogelezang and Holtslag, 1996; Liang and Liu, 2010), we divide the ABLs into three types: stable boundary layer (SBL), near-neutral boundary layer (NBL), and convective boundary layer (CBL), corresponding with three different stability states near the surface. We first use $SH$ to diagnose the ABL regime types. The specific classification formula is presented below:

$$\begin{cases} SH > +\delta & \text{for CBL} \\ SH < -\delta & \text{for SBL}, (1) \\ else & \text{for NBL} \end{cases}$$

[revised manuscript text omitted]

Ln 178: theta is used here as measure for stratification. However, above you mention that the PBL driven by turbulence in cloud is an ABLH important archetype. Is it not more appropriate to use a temperature metric that is conserved in moist conditions like the liquid water potential temperature? Please show that this choice does not affect your conclusions!

Response: Thank you very much for your helpful suggestion. We have replaced $\theta$ with the equivalent potential temperature $\theta_E$. As mentioned above, the types of ABLs are first diagnosed by *SH* data. Only when the *SH* is missing are the sounding profiles classified by the difference of $\theta_E$ between the 100 and 50 m heights. We also checked the impact of using $\theta_E$ on regime classification. For the $\theta$ criterion, we obtain 452 SBLs, 240 NBLs, 272 CBLs. For the $\theta_E$ criterion, we obtain 442 SBLs, 249 NBLs, 273 CBLs. This slight difference does not affect our conclusions.

Ln 180: delta_s is chosen to be 0.2 K. Please relate link this to the measurement accuracy of the sounding. In my view even for a routine AWS the measurement

uncertainty is about 0.3K when it includes also representativeness uncertainty.

Response: Thank you very much for your helpful comment. According to the description in Liu and Liang (2010), $\delta_s$ is the $\theta$ increment for the minimum strength of the stable (inversion) layer. The value of $\delta_s$ would be set to zero for idealized cases but in practice is specified as a small positive value, and this value depends on the surface characteristics as well as inherent uncertainties or noise in the measurements. For profiles over ocean and ice, this threshold has been empirically defined to be 0.2 K. While, as described in the measurement data report, the measurement uncertainty is exactly 0.3 K. Considering the uncertainty of this criterion, we have replaced it with the *SH* data to determine the ABL regime types, which follows Steeneveld et al. (2007b). Only when the *SH* is missing is the $\theta_E$ criterion and threshold proposed by Liu and Liang (2010) used for determining ABL type. For this comment, the corresponding change is given in our revised manuscript as follows:

**3.1 ABL regime classification and ABLH determination**

The ABLH determination method starts with the classification of ABL regimes. Based on previous studies (e.g., Vogelezang and Holtslag, 1996; Liang and Liu, 2010), we divide the ABLs into three types: stable boundary layer (SBL), near-neutral boundary layer (NBL), and convective boundary layer (CBL), corresponding with three different stability states near the surface. We first use *SH* to diagnose the ABL regime types. The specific classification formula is presented below:

$$\begin{cases} SH > +\delta & \text{for CBL} \\ SH < -\delta & \text{for SBL} \\ else & \text{for NBL} \end{cases}, (1)$$

where $\delta$ is the critical value that is specified as 2 W m$^{-2}$, following Steeneveld et al. (2007b). If corresponding *SH* data are unavailable, the difference of equivalent potential temperature between the 100 and 50 m heights ($\theta_{Egrad}$) derived from sounding profiles is used to determine the ABL types. Specifically, if the $\theta_{Egrad}$>0.2 K, the ABL is identified as SBL; if the $\theta_{Egrad}$<-0.2 K, the ABL is identified as CBL; and other profiles are labeled as NBLs, roughly following Liu and Liang (2010).

Ln 226: which an air parcel rising adiabatically from the surface becomes neutrally buoyant... Has an temperature excess been added to the surface parcel and if so with which value?

Response: Thank you very much for your helpful comment. According to Liu and Liang (2010) and the PBLH VAP data report, a temperature excess of 0.1 K has been added. Actually, the ABLH estimates based on the Liu-Liang algorithm is provided by the data product, and we directly use them for comparison. We have added more relevant information into our revised manuscript as follows:

The Liu-Liang algorithm determines ABLH based on potential temperature and wind speed. For CBL regimes, the definition of ABLH is the height at "which an air parcel rising adiabatically from the surface becomes neutrally buoyant", and the temperature excess value is 0.1 K.

Ln 227+228: two different estimates of the SBL height are obtained based on stability criteria and wind shear criteria, respectively. Please elaborate in more detail how it has been done, in this way we cannot evaluate the procedure is appropriate.

Response: Thank you very much for your helpful comment. According to Liu and Liang (2010) and the ABLH VAP data report, the stability criteria are to find the lowest level, $k$, at which the $\theta_{Egrad}$ reaches a minimum and meets either of the following two conditions:

$$\begin{cases} \theta_{Egrad\,k} - \theta_{Egrad\,k-1} < \text{-40 K/km} \\ \theta_{Egrad\,k+1} < 0.5 \text{ K/km}, \theta_{Egrad\,k+2} < 0.5 \text{ K/km} \end{cases}'$$

where the subscripts ($k$, $k$-1, $k$+1, and $k$+2) represent the $\theta_{Egrad}$ at corresponding levels.

For wind shear, the ABLH is defined as the height where the wind speed reaches a maximum that is at least 2 m/s stronger than the layers immediately above and below while decreasing monotonically toward the surface (i.e., a low-level jet). The final ABLH is defined as the lower of the two heights. We have added more information into our revised manuscript as following:

For SBL regimes, two different estimates of the ABLH are obtained, if possible, based on stability criteria and wind shear criteria, respectively. For stability, the ABLH is defined as the lowest level, $k$, at which the $\theta_{Egrad}$ reaches a minimum and meets either of the following two conditions:

$$\begin{cases} \theta_{Egrad\,k} - \theta_{Egrad\,k-1} < \text{-40 K/km} \\ \theta_{Egrad\,k+1} < 0.5 \text{ K/km}, \theta_{Egrad\,k+2} < 0.5 \text{ K/km} \end{cases}, (2)$$

where the subscripts ($k$, $k$-1, $k$+1, and $k$+2) represent the $\theta_{Egrad}$ at corresponding levels.

For wind shear, the ABLH is defined as the height where the wind speed reaches a maximum that is at least 2 m/s stronger than the layers immediately above and below while decreasing monotonically toward the surface (i.e., a low-level jet). The final ABLH is defined as the lower of the two heights.

Ln 239: dimensional number. It is a dimensionLESS number, of course!

Response: Thank you very much for pointing this out. It is a typo error and we have corrected it.

Ln 240-246: the paper ignores here the knowledge that was developed in Vogelezang and Holtslag, which was by the way cited, that a better score for the ABLH can be obtained if Equation 2 is not considered from the surface parcel, but a parcel at somewhat above the surface. Hence I feel the latest knowledge is not taken into account here.

Response: Thank you very much for your helpful comment. As mentioned above, we have considered the $Ri$ formula proposed by Vogelezang and Holtslag (1996) into our revised manuscript, and use it in the ABL algorithm for clear-sky conditions. The corresponding changes are given in our revised manuscript as follows:

**3.3 An improved $Ri$ algorithm considering the cloud effect**

As a traditional $Ri_b$ formula, Eq. (3) may break down in cases of ABLs with relatively high wind speed and upper-level stratification due to the overestimation of shear production (Kim and Mahrt, 1992). Vogelezang and Holtslag (1996) proposed the finite-difference $Ri$ formula, which is expressed as:

$$Ri_F = \frac{(g/\theta_{vs})(\theta_{vh} - \theta_{vs})(h - z_s)}{(u_h - u_s)^2 + (v_h - v_s)^2 + bu_*^2}, (6)$$

where $z_s$ is the lower boundary for the ABL, $\theta_{vs}$, $u_s$, and $v_s$ are the $\theta_v$ and wind components at the height $z_s$, respectively, b is an empirical coefficient, and $u_*$ is the surface friction velocity. $Ri_F$ is considered for a parcel located somewhat above the surface to avoid the above problem, and $u_*$ is also taken into account to avoid underestimation in the situation of a uniform wind profile in the upper layer. Here, we use $Ri_F$ for clear-sky profiles and take $z_s$ and $b$ values as 40 m and 100, respectively, according to Zhang et al. (2020).

Ln 254: $Bias$ is the absolute bias; $SEE$ is the standard error. I object against the term bias here. Bias can be either positive or negative, but your formula for bias cannot, so you use the MAE, mean absolute error. Idem for SEE, it is the standard deviation of the error, not the standard deviation of the ABLH.

Ln 257: note that Steeneveld et al. (2007) used the median of the absolute error is evaluation metric in a similar type of study. This is helpful to avoid that the error statistics are determined strongly by one or two outliers. Please consider this as well.
Figure 3: it is unclear whether the error statistics in the left upper corner relate to the CBL or SBL data. It would be interesting to have the statistics for both classes, to underline the score for SBL is much poorer.

Response: Thank you very much for pointing this out. We have revised the formula of *Bias* that can be positive or negative, and replaced other statistical measures with median of the absolute error (*MEAE*). For Fig. 3, we have calculated the error statistics for SBL, NBL, CBL, and cloudy conditions, and listed them in Table 1. According to the error statistics, the Liu-Liang algorithm and the Heffter algorithm perform poorly in determining SBL height, especially the Liu-Liang algorithm. For this comment, the corresponding changes are given in our revised manuscript as follows:

(1) Changes related to the description of statistical measures

To quantitatively evaluate the performance of each automatic algorithm, we introduce the correlation coefficient $R$ and two other statistical measures: the *Bias* and the median absolute error (*MEAE*; Steeneveld et al., 2007a). The formulas are as follows:

$$Bias = \frac{2}{n} \sum_{i=1}^{n} \frac{H_{auto}\text{-}H_{obs}}{H_{auto}\text{+}H_{obs}}, (4)$$

$$MEAE = \text{median}(|H_{auto}\text{-}H_{obs}|), (5)$$

where $H_{auto}$ is the ABLH obtained by the automated algorithm; $H_{obs}$ is the ABLH manually determined; $n$ is the number of valid sounding profile samples. According to the definitions of these statistical measures, larger $R$ and smaller *Bias* and MEAE mean a better performance of the automated algorithm.

(2) Changes related to Figure 3

Figure 3 presents the comparisons of estimated ABLHs with the manually-labeled ABLHs, and the associated statistical measures are given in Table 1. The results show that the $Ri_b$ algorithm with $Ri_{bc}$ of 0.25 performs best overall, and particularly for SBL cases. The performance of the $Ri_b$ algorithm with $Ri_{bc}$ of 0.5 is poorer than that of the $Ri_b$ algorithm with $Ri_{bc}$ of 0.25, with overestimations of ABLHs in general, and larger errors with lower correlation coefficients for all types of ABLs. The Heffter algorithm performs well in cases of high ABLH and particularly for cloudy and CBL cases, but does significantly overestimate ABLH in a large number of cases as shown in the Fig. 3c subgraph. This is attributed to the determination criterion of the Heffter algorithm, i.e., ABLHs are determined by inversion layers, which means that large errors occur when the inversion layer is higher than the mixed layer. Additionally, while the Heffter performance in many of the ABL conditions is only marginally worse statistically than the $Ri_b$ algorithm with $Ri_{bc}$ of 0.25, its correlations are notably worse for SBL and NBL cases. The performance of the Liu-Liang algorithm is generally poorer than the other algorithms, particularly for correlation coefficient, which is probably due to the impact of noise in the lower ABLH profiles and unsuitable parameters in the algorithm. In summary, the $Ri_b$ algorithm is reliable over the Arctic Ocean and performs better than other algorithms, and this result agrees with Jozef et al. (2022). Furthermore, we will explore ways to improve the $Ri_b$ algorithm to make it more suitable for cloudy and convective conditions.

[Figure]

Figure 3 Comparisons of the ABLHs determined from radiosonde profiles using the bulk Richardson number ($Ri_b$) algorithm with the critical values ($Ri_{bc}$) of (a) 0.25 and (b) 0.5, (c) the Heffter algorithm, and (d) the Liu-Liang algorithm with the manually-identified "observed" ABLHs. The blue, yellow, and red colors indicate regime types of SBL, NBL, and CBL, respectively. The "x" signs indicate the Cloudy ABLs. The case numbers ($N$) and correlation coefficients ($R$) are given in each panel. The subgraph in (c) denotes all data points ranging from 0 to 3.5 km.

Figure 3c and d: I do not understand why the H_obs is different for the SBL and the CBL for the two panels. Please explain, the filtering was done on the observation, wasn't it? Not on the selected algorithm. Also add the number of samples in the block with error statistics.

Response: Thank you very much for pointing this out. We realize that the Fig. 3 is unclear. Actually, the $H_{obs}$ is the same in all panels and the filtering was in fact done on the observations. The data range of Fig. 3c in the original manuscript is from 0–3.5 km due to the severe ABLH overestimation by the Heffter algorithm, and the axis range is

thus different from that of other panels. Therefore, we unify the axis ranges of all panels to avoid misunderstandings, and add a subgraph in Fig. 3c to denote all data points. For this comment, the corresponding changes are given in our revised manuscript as follows:

[Figure]

Figure 3 Comparisons of the ABLHs determined from radiosonde profiles using the bulk Richardson number ($Ri_b$) algorithm with the critical values ($Ri_{bc}$) of (a) 0.25 and (b) 0.5, (c) the Heffter algorithm, and (d) the Liu-Liang algorithm with the manually-identified "observed" ABLHs. The blue, yellow, and red colors indicate regime types of SBL, NBL, and CBL, respectively. The "x" signs indicate the Cloudy ABLs. The case numbers (N) and correlation coefficients (R) are given in each panel. The subgraph in (c) denotes all data points ranging from 0 to 3.5 km.

Ln 296: Note again that VH96 do use a different definition of Ri.

Response: Thank you very much for pointing this out. We have considered it into our revised manuscript as mentioned above.

Ln 302: This result is distinct from that of Jozef et al. (2022). Add how it is distinct....?

Ln 303: might be that ... different... ->Better to figure that out!!! It is related to the key of this paper.

Response: Thank you very much for your helpful comment. We have added the analysis of different $Ri$ formulas and $Ri_c$ values, and the $Ri$ formula used in Jozef et al. (2022) is also included. Jozef et al. (2022) calculates the $Ri$ over a rolling 30 m altitude range, and uses the $Ri_c$ value of 0.75. The method of calculating $Ri$ over a rolling 30 m range causes dramatic variation within the ABL, as seen in Fig. 5. Thus, for this $Ri$ definition, a large $Ri_c$ value is required to avoid the noise. The corresponding changes are given in our revised manuscript as follows:

Since some other studies have proposed different $Ri_c$ values for MOSAiC (e.g., Jozef et al., 2022; Barten et al., 2023; Akansu et al., 2023), we will discuss the difference in $Ri_c$ values here. The first thing to make clear is that these studies use different formulas to obtain $Ri$ profiles. Barten et al. (2023) and Akansu et al. (2023) both use the traditional $Ri_b$ algorithm based on Eq. (3), while they used $Ri_c$ values of 0.4 and 0.12, respectively. This difference was likely caused by the different methods to manually derive their reference ABLH data sets. Jozef et al. (2022) calculates the $Ri$ over a rolling 30 m altitude range, labeled as $Ri_r$, and the criterion is modified to require four consecutive data points to be above the $Ri_c$ of 0.75. In our study, we use $Ri_F$ proposed by Vogelezang and Holtslag (1996) for clear-sky conditions, and $Ri_m$ for cloudy conditions. Based on the results presented here, it is apparent that this more complex approach improves the error statistics relative to approaches based on Eq. (3), regardless of $Ri_c$. In addition, some of the differences may also related to authors using different data sets or time periods. For instance, Akansu et al. (2023) primarily used sounding data based on tether balloon for a specific sub-period of MOSAiC, and Jozef et al. (2022) used radiosondes from when they had concurrent UAV observations. The data used in our study are based on merged sounding-tower product, as mentioned above.

To further explore the differences among the four different approaches, we examine one SBL and CBL case. For a clear-sky SBL case (Fig. 5 a, b), the approaches from Akansu et al., Jozef et al. (2022), and this study all agree closely with the manual ABLH, while the Barten et al. approach results in a significant overestimation. For a cloudy-sky CBL case (Fig. 5 c d), the approach from this study agrees with the manual ABLH, while the approach from Barten et al. overestimates the ABLH by about 30 m, and the approaches from Akansu et al. and Jozef et al. (2022) underestimate the ABLH by 130 m and 230 m, respectively. These results further demonstrate how $Ri_c$ depends on the choice of $Ri$ formula. Moreover, $Ri_c$ is not analytically derived from basic physical principles (Zilitinkevich et al. 2007), and the concept of $Ri_c$ is challenged by non-steady regimes (Zilitinkevich and Baklanov, 2002) and the hysteresis phenomenon (Banta et al., 2003; Tjernström et al., 2009). Therefore, an objective $Ri_c$ does not exist. Rather, it

is empirically used as an algorithmic parameter to simply derive the ABLH.

[Figure]

Figure 5 Vertical profiles of (left) $\theta_E$ and wind speed, and (right) $Ri$ based on different formulas at (a–b) 25 November 2019, 22:58 UTC and (c–d) 17 December 2019, 16:58 UTC. Boundary layers at the two times represent a clear-sky SBL and a cloudy-sky CBL respectively. The black dashed horizontal lines denote the manually-identified ABLH, and the gray solid vertical lines denote the different $Ri_c$ values, including 0.12, 0.35, 0.4, and 0.75. The gray shading in (c) denotes the cloud layer.

Ln 328: from 13 April through to 24 May 2020. In this period, the convectively thermal structure contributes to ABLH reaching over 610 m for about 6 days, with the maximum ABLH of 1152 m: This is the period with a warm intrusion from the south, so the PBL height is likely strongly governed by the advection of warm air, its turbulent kinetic energy, and its stratification. Equation 2 was not developed for such conditions, so it is fair to evaluate it as such?

Response: Thank you very much for your helpful comment. We checked the ABL cases in this period, and the comparisons of ABLH respectively estimated by the traditional and improved $Ri$ algorithms with $H_{obs}$ are presented in Fig. R1. There are 108 ABL cases in this period. We find that the $Ri$ algorithm based on Eq. (3) surely cannot

determine ABLH well, while the improved $Ri$ algorithm significantly corrects for errors in ABLH estimation, by using the $Ri_F$ proposed by VM96 and taking the cloud effect into account.

[Figure]

Figure R1 The Comparisons of the selected ABLHs determined by the (a) bulk Richardson number ($Ri_b$) algorithms with the critical values ($Ri_{bc}$) of 0.25 and (b) the improved $Ri$ algorithm with observed ABLHs. The blue, yellow, and red dots indicate regime types of SBL, NBL, and CBL, respectively. The case number ($N$), correlation coefficient ($R$), $Bias$, and median of the absolute error ($MEAE$) are given in each panel.

Ln 366: I am little surprised that the theta_E appears here in the analysis, while it is not reasoned why we step over from theta to theta_E. I agree that theta_E analysis is valuable, but should theta_E not have been applied to Equation 2?

Response: Thank you very much for your helpful comment. Actually, the Eq. (2) and corresponding ABLH estimations are provided by the ABLH VAP data product, so we have used it directly for comparison analysis. In our improved algorithm, we use the moist Richardson number to take the cloud effect into account. Except for Eq. (2), we have replaced $\theta$ with $\theta_E$ in the rest of our manuscript for consistency.

Figure 7: Add in the legend whether these are the monthly averages of the soundings from 5:00, or 11:00, or 17:00, or 23:00, or all mixed together. It is better to stick to one time slot to avoid that the effects of the diurnal cycle in the summer months are mixed away.

Response: Thank you very much for your helpful comment. In our study, all soundings from 5:00, or 11:00, or 17:00, or 23:00 are used to calculate the monthly averages. In

order to check the impact of the diurnal cycle, we also calculate respective monthly profiles based on 5:00, 11:00, 17:00 and 23:00. The results are presented in Fig. (R2–R5). We find that the monthly profiles based on the four individual hours of a day do not differ much, which suggests that the diurnal cycle does not have a significant effect on the ABL thermal structure. Therefore, we continue to use the monthly profiles based on all soundings, and add the relevant statement at the end of the analysis as follows:

In addition, we examined the potential implications of the diurnal cycle on the ABL thermal structure. Monthly profiles based on different moments of a day were found to show little variability (not shown), such that the impact of the diurnal cycle is minimal.

[Figure]

Figure R2 Median profiles of equivalent potential temperature throughout the MOSAiC year are divided into (a), (b), and (c), based on sounding data from 5:00.

[Figure]

Figure R3 Median profiles of equivalent potential temperature throughout the MOSAiC year are divided into (a), (b), and (c), based on sounding data from 11:00.

[Figure]

Figure R4 Median profiles of equivalent potential temperature throughout the MOSAiC year are divided into (a), (b), and (c), based on sounding data from 17:00.

[Figure]

Figure R5 Median profiles of equivalent potential temperature throughout the MOSAiC year are divided into (a), (b), and (c), based on sounding data from 23:00.

Ln 395: temperature gradient. Better to use (equivalent) potential temperature gradient to remain consistent with the above.

Response: Thank you very much for your helpful suggestion. We have replaced the temperature gradient with equivalent potential temperature gradient $\theta_{Egrad}$ for consistency. The corresponding changes are given in our revised manuscript as follows:

To further explore the relations between surface conditions and the ABLH, we evaluate the correlations between the ABLH and three near-surface meteorological and turbulence parameters during the MOSAiC period, including the near-surface equivalent potential temperature gradient ($\theta_{Egrad} = \theta_{E\ 10m} - \theta_{E\ 2m}$), friction velocity

($u_*$), and TKE dissipation rate ($\varepsilon$). The results are shown in Fig. 11. Generally, the near-surface buoyancy and shear effects both modulate these variables. In Fig. 11a, the ABLH distribution for negative $\theta_{Egrad}$ has a wide range from the lowest level to above 1 km. As $\theta_{Egrad}$ becomes positive and increases, the ABLH distribution rapidly narrows to below 200 m. In general, positive $\theta_{Egrad}$ means a stably stratified ABL and surface-based temperature inversion, both of which lead to low ABLH, and negative $\theta_{Egrad}$ means that atmospheric stability near the surface is near-neutral or convective, which is necessary for ABL development.

[Figure]

Figure 11 The ABLHs and bin-averaged values for (a) equivalent potential temperature gradient, $\theta_{Egrad}$ (K), (b) friction velocity, $u_*$ (m s$^{-1}$), and (c) turbulent kinetic energy dissipation rate, $\varepsilon$ (m$^2$ s$^{-3}$). The average bins for $\theta_{Egrad}$, $u_*$, and $\varepsilon$ logarithm are 0.2 K, 0.05 m s$^{-1}$, and 0.5 m$^2$ s$^{-3}$, respectively. The correlation coefficient $R$ is given in (b), which is statistically significant ($p < 0.05$). The dashed vertical lines indicate the thresholds of (a) $\theta_{Egrad}$ = 0 K and (c) $\varepsilon$ = 5×10$^{-5}$ m$^2$ s$^{-3}$.

Ln 396: u*, * should be subscripted (twice). And in the rest of the manuscript.

Response: Thank you very much for pointing this out. We have corrected relevant expressions.

Figure 8a and c: The R value in the plot is an estimate for the LINEAR correlation between the two variables, but obviously the relation is not linear. So better to remove it, or first do a transformation on the data such that the relation between them becomes linear.

Response: Thank you very much for pointing this out. We have removed the correlation analysis in Fig. 11a and c, retaining only in Fig. 11b. The corresponding change is given in our revised manuscript as following:

[Figure]

Figure 11 The ABLHs and bin-averaged values for (a) equivalent potential temperature gradient, $\theta_{Egrad}$ (K), (b) friction velocity, $u_*$ (m s$^{-1}$), and (c) turbulent kinetic energy dissipation rate, $\varepsilon$ (m$^2$ s$^{-3}$). The average bins for $\theta_{Egrad}$, $u_*$, and $\varepsilon$ logarithm are 0.2 K, 0.05 m s$^{-1}$, and 0.5 m$^2$ s$^{-3}$, respectively. The correlation coefficient $R$ is given in (b), which is statistically significant ($p < 0.05$). The dashed vertical lines indicate the thresholds of (a) $\theta_{Egrad}$ = 0 K and (c) $\varepsilon = 5 \times 10^{-5}$ m$^2$ s$^{-3}$.

Figure 8b: it is interesting to note that the ABLH is about 700xu*, which was also found/discussed in Vogelezang and Holtslag (1996) and Steeneveld et al. (2007). Both studies also explore ABLH=10u*/N as ABLH estimate, it would be interesting to be tested here as well.

Response: Thank you very much for your helpful suggestion. As mentioned above, we have added the tests for SBL formulas, including $h_E = \alpha \frac{u_*}{N}$. The results indicate that the best-fit $\alpha$ value is 20, not the typical value of 10, but this is in agreement with Overland and Davidson (1992), whose data also come from the ABL over sea ice. Thus, we attribute this result to the unique free-flow stability or other potential mechanisms of ABL development in the Arctic atmosphere. The corresponding changes are given in our revised manuscript as follows:

The free-flow stability (characterized by the free-flow Brunt-Väisälä frequency, $N$) can affect the ABLH (Zilitinkevich et al., 2002; Zilitinkevich and Baklanov, 2002; Zilitinkevich and Esau, 2002, 2003), and therefore is also examined here. Based on the buoyancy flux at the surface ($B_s$) and $N$, the NBLs and SBLs can be further divided into four types: the truly neutral (TN, $B_s$ = 0 and $N$ = 0), the conventionally neutral (CN, $B_s$ = 0 and $N > 0$), the nocturnal stable (NS, $B_s < 0$ and $N$ = 0), and the long-lived stable boundary layer (LS, $B_s < 0$ and $N > 0$). According to Zilitinkevich and Baklanov (2002), we calculate the $N$ and $B_s$ and reclassify the SBLs and NBLs. We find that the percentages of $N > 0.015$ in SBLs and NBLs are 89 % and 80 %, which indicates that LS and CN types dominate the stable and neutral conditions for MOSAiC, respectively. Since only 80 TN cases were identified, these are deemed to be too few for additional analysis of this type. Zilitinkevich and Esau (2003) gave ABLH equations relevant to

each ABL type as:

$$h_E=\begin{cases} C_N u_* |fN|^{-1/2} & \text{(Pollard et al., 1973)} & \text{for CN ABL, (10)} \\ C_S u_*^2 |fB_s|^{-1/2} & \text{(Zilitinkevich, 1972)} & \text{for NS and LS ABL, (11)} \end{cases}$$

where $h_E$ is the equilibrium ABLH, $f$ is the Coriolis parameter, and $C_N$ and $C_S$ are empirical coefficients. In addition, Vogelezang and Holtslag (1996) and Steeneveld et al. (2007a) also explore a $h_E$ equation without taking into account $f$ explicitly, expressed as:

$$h_E=C_i \frac{u_*}{N} \quad \text{for all SBL and NBL, (12)}$$

where $C_i$ is an empirical coefficient. Here we select the CN, NS, and LS ABLH dataset, and fit the data with the corresponding expressions in Eq. (10–12) to obtain the empirical coefficients, and the results are presented in Fig. 12. All three expressions tend to well represent the ABLHs, with significant correlation coefficients. The empirical coefficients $C_N$ and $C_S$ are 1.7 and 0.4, respectively, which are close to the typical values determined through large-eddy simulations (Zilitinkevich, 2012). The coefficient $C_i = 20$ in Fig. 12c is double the typical value of 10 (Vogelezang and Holtslag, 1996), but agrees with the results reported by Overland and Davidson (1992) for the ABL over sea ice. The difference in $C_i$ may be attributed to the unique free-flow stability or other potential mechanisms of ABL development in the Arctic atmosphere.

In summary, near-surface conditions and free-flow stability play a key role in ABL development and are also an indicator, in that one can roughly determine the development state of the whole ABL from these basic variables.

[Figure]

Figure 12 The ABLHs versus three expressions in Eq. (10–12). The empirical coefficients $C_N$, $C_S$, and $C_i$ are given in (a), (b), and (c), respectively, and represent the slope of the best fit line (black line). The correlation coefficient $R$ is given in each panel, which is statistically significant ($p < 0.05$).

Fig 9, caption: wind speed -> horizontal wind speed

Response: Revised as suggested.

Fig 10: Figure 10 Similar to Fig. 9, but the period is from 15 July 2020 to 30 August 2020. Legend is likely wrong since the x axis goes surely beyond September 1st.

Response: Thank you very much for pointing this out. We realized that the axis range of the Figure is beyond what we expect, and we have corrected it. The corresponding change is given in our revised manuscript as follows:

[Figure]

Figure 14 Similar to Fig. 13, but the period is from 15 July 2020 to 30 August 2020.

---

## Author Response (AR2)

Dear Thijs,

Thank you very much for handling our manuscript (egusphere-2023-347). We have substantially revised the manuscript by addressing all the reviewers' comments. The revisions in the manuscript and the reply to the comments are marked in blue. Thank you very much.

Sincerely,

Changwei Liu (liuchw8@mail.sysu.edu.cn)
On behalf of all the authors
* * *
**Anonymous Referee #1**

Summary: The authors have adequately addressed my previous comments. I also find that the quality of the manuscript, including the language, has been significantly improved. I only have a few additional minor comments.

Response: Thank you very much for your time and effort in reviewing our manuscript. We have revised the manuscript accordingly. The revisions in the manuscript and the reply to the comments are marked in blue.

**Minor comments:**

1, I would suggest add "the" in the title, "Characteristics of the atmospheric boundary layer height over the Arctic Ocean during MOSAiC"
Response: Revised as suggested.

2, line 20-22: this is not very accurate. The summer temperature inversion is intensified because of warm air advection with surface temperature constrained by melting, but by surface melting.
Response: Thank you very much for pointing this out. We have revised the sentence as "Temperature inversions in the winter and summer are intensified by seasonal radiative cooling and warm air advection with surface temperature constrained by melting, respectively, leading to the low ABLH at these times."

3, line 22-23: Although friction velocity is a surface variable, it is actually not only related to surface characteristics. I would not use it as representing near-surface conditions. Just say that ABL variation is correlated with friction velocity and TKE dissipation rate. Also, the reviewer needs to elaborate on "these basic variables" or reword this sentence by avoiding such vague language.
Response: Thank you very much for pointing this out. We have revised the sentence as

"Meteorological and turbulence variables also play a significant role in ABLH variation, including near-surface potential temperature gradient, friction velocity, and TKE dissipation rate."

4, line 135: suggest remove "strongly"
Response: Revised as suggested.

5, line 137: suggest remove "predominately"
Response: Revised as suggested.

6, line 200-201: it's still confusing to me in terms of "multiple profiles". I think it should be "The gray dashed horizontal lines in each panel denote the atmospheric boundary-layer height (ABLH) estimates based only on the profile shown in that panel (e.g., in panel the gray dashed horizontal line is determined only using the profile of theta_E)".
Response: Revised as suggested.

7, line 331: "regardless of Ric" seems too strong. I don't see a proof of this statement.
Response: Thank you very much for pointing this out. We have removed the relevant statement.

8, line 452-453: it should be "could have been potentially impacted by more open-surface water conditions"
Response: Revised as suggested.

9, line 499-501: this sentence is not very accurate. Epsilon is the dissipation rate. It can only indicate the rate at which the TKE is changing, not the magnitude of TKE itself.
Response: Thank you very much for pointing this out. We have revised the sentence as "The $\varepsilon$ indicates the rate at which the TKE is changing, and the high value of $\varepsilon$ means well-developed turbulence"

10, line 529: suggest change "an indicator" to "indicators"
Response: Revised as suggested.

11, line 545: suggest remove "the unique characteristics of" and "in detail"
Response: Revised as suggested.

12, line 590: "that" should be removed.
Response: Revised as suggested.
* * *
**Anonymous Referee #2**

Summary: The authors have done a good and thorough work to take into account the raised concerns. The manuscript can be published pending some editorial items:

Response: Thank you very much for your time and effort in reviewing our manuscript. We have revised the manuscript accordingly. The revisions in the manuscript and the reply to the comments are marked in blue.

Ln 222: K/km -> km-1
Response: Revised as suggested.

Ln 224: m/s -> ms-1
Response: Revised as suggested.

Table 1: the bias is missing a unit in the header. Also I do not understand the bias values, since I expect them to be in the same order of magnitude as for MEAE. Is this really the bias in meters?
Response: Thank you for your helpful comment. Actually, the *bias* in our manuscript is defined as a dimensionless metric. According to Eq. (4), the range of bias can be [-2, 2], and algorithms perform well as bias is close to 0. This bias definition helps avoid the influence of ABLH ($H_{obs}$) itself on the bias values. For this comment, we have added a "dimensionless" into the *bias* metric description.

Equation 7: I think each term in the denominator should be in brackets and then squared (i.e. not the gradients of u^2 and v^2.
Response: Revised as suggested.

Ln 365: .... the Obukhov length. Please add "at the surface" behind "length"
Response: Revised as suggested.

Ln 458: remove space after first )
Response: Revised as suggested.

Ln 483: ....divided into (a), (b), and (c). Please reword. Just mention panel a represents Oct-Jan, panel b Feb-May and panel c) Jun -Sep.
Response: Revised as suggested.

Ln 661: remove space in "L ow"
Response: Revised as suggested.

Ln 684: 'new Arctic.'. Remove dot behind Arctic.
Response: Revised as suggested.